# Dimorphic neural network architecture prioritizes sexual-related behaviors in male *Caenorhabditis elegans*

Xuebin Wang[1,2,3†], Hanzhang Liu[1,2,3†], Wenjing Yang[4†‡], Jingxuan Yang[5], Xuehong Sun[5], Qiuhan Liu[4], Ying Zhu[4], Yinghao Sun[1,2,3], Chunxiuzi Liu[1,2,3], Guiyuan Shi[1,2], Qiang Liu[6], Ke Zhang[1,2], Zengru Di[1,2], Wenxing Yang[4], He Liu[1,2]*

[1]Department of Systems Science, Faculty of Arts and Sciences, Beijing Normal University, Zhuhai, China; [2]International Academic Center of Complex Systems, Beijing Normal University, Zhuhai, China; [3]School of Systems Science, Beijing Normal University, Beijing, China; [4]Huitong College, Beijing Normal University, Zhuhai, China; [5]Department of Physiology, West China School of Basic Medical Sciences and Forensic Medicine, Sichuan University, Chengdu, China; [6]Department of Neuroscience, City University of Hong Kong, Hong Kong, China

**\*For correspondence:**
heliu@bnu.edu.cn

†These authors contributed equally to this work

**Present address:** ‡Department of Organismic and Evolutionary Biology, Harvard University, Cambridge, United States

### eLife Assessment

This study presents **useful** findings on the differences between male and hermaphrodite *C. elegans* connectomes and how they may result in changes in locomotory behavioral outputs. However, the study appears **incomplete** with respect to the relationship between sex-specific AVA wiring and male mate-finding. Another area of concern is that the analysis does not consider animal-to-animal variability in the wiring when attempting to identify significant differences between the male and hermaphrodite.

**Abstract** Neural network architecture determines its functional output. However, the detailed mechanisms are not well characterized. In this study, we focused on the neural network architectures of male and hermaphrodite *Caenorhabditis elegans* and the association with sexually dimorphic behaviors. We applied graph theory and computational neuroscience methods to systematically discern the features of these two neural networks. Our findings revealed that a small percentage of sexual-specific neurons exerted dominance throughout the entire male neural network, suggesting males prioritized sexual-related behavior outputs. Based on the structural and dynamical characteristics of two complete neural networks, sub-networks containing sex-specific neurons and their immediate neighbors, or sub-networks exclusively comprising sex-shared neurons, we predicted dimorphic behavioral outcomes for males and hermaphrodites. To verify the prediction, we performed behavioral and calcium imaging experiments and dissected a circuit that is specific for the increased spontaneous local search in males for mate-searching. Our research sheds light on the neural circuits that underlie sexually dimorphic behaviors in *C. elegans* and provides significant insights into the interconnected relationship between network architecture and functional outcomes at the whole-brain level.

## Introduction

The concept that neural network architecture plays a crucial role in determining functional output has gained widespread acceptance in the neuroscience field (*Luo, 2021*), and it has also been applied in artificial intelligence, where optimizing network performance involves modifying the network architecture (*Prescott and Wilson, 2023*). The same number of neurons form specialized architectures that may result in distinct functions; for example, neurons may execute continuous topographic mapping in vision system or discrete parallel processing in olfactory circuit (*Luo, 2021*). The convergent-divergent feature in the *Drosophila* olfactory circuit facilitates a wide range of odor representations and supports olfactory memory (*Benton, 2022*; *Boto et al., 2020*; *Endo et al., 2020*; *Jeanne and Wilson, 2015*; *Masse et al., 2009*). The modified characteristics of the engram cell at both cellular and circuitry levels elucidate the underlying mechanisms of learning and memory (*Josselyn and Tonegawa, 2020*), for example, AMPA receptors recycling on the membrane mediates the synaptic strength to regulate memory formation and forgetting (*Awasthi et al., 2019*; *Guskjolen, 2016*). However, due to the absence of connectome data, limited research has been directed towards exploring the relationship between neural network architecture and the behavioral functional output at the whole-brain level.

Sexually dimorphic behavior provides an intriguing platform to study the relationship between neural network architectures and behavioral outcomes. Sexually dimorphic behaviors are generally categorized into two types: those with qualitative differences, observed in only one sex, and those with quantitative differences, displayed by both sexes but at varying levels (*Kelley, 1988*). Some animals exhibit sex-specific behaviors. For example, male satin bowerbirds construct intricate bowers and engage in elaborate courtship displays nearby to attract females. The female assesses the quality of the bower and the male's performance for mate selection (*Coleman et al., 2004*). Studies have discovered the molecular and circuitry mechanisms of sex-specific behaviors in some cases. For instance, the male-specific isoform of Fruitless, Fru(M), in *Drosophila* plays an essential role in regulating male courtship behavior. Interestingly, females expressing Fru(M) also exhibit courtship behavior (*Demir and Dickson, 2005*). Male-specific P1 interneuron drives male courtship initiation and motivation (*Bath et al., 2014*; *von Philipsborn et al., 2011*; *Zhang et al., 2021*). Pregnant female mice actively participate in nesting behaviors to create a suitable environment for offspring. Using brain-wide immediate early gene mapping, *Topilko et al., 2022* discovered that the Edinger–Westphal (EW) nucleus plays an important role in regulating nesting behavior. Males tend to exhibit more frequent or intense aggressive behaviors than females (*Pandolfi et al., 2021*). However, reproductive state significantly influences female attack behavior, as evidenced by lactating female mice displaying maternal aggressiveness to protect the offspring (*Liu et al., 2022a*). Male aggression is mediated by the progesterone receptor (PR) expressing neurons located in the ventromedial hypothalamus (*Yang et al., 2013*), and artificially activating these neurons triggers male aggression (*Yang et al., 2017*). Estrogen receptor-α (Esr1)-positive neurons in female ventromedial hypothalamus are essential for lactation-related aggression (*Hashikawa et al., 2017*; *Liu et al., 2022a*).

*Caenorhabditis elegans* has two sexes, male and hermaphrodite. Numerous studies have unveiled sexual dimorphism at gene expression, neural circuitry, and behavioral levels (*Burkhardt et al., 2023*; *Cahoon et al., 2023*; *Emmons, 2018*; *Ma et al., 2024*; *Haque et al., 2024*). Males are attracted to hermaphrodite pheromones, through dimorphic expression of pheromone receptors in male-specific neurons and some sex-shared sensory neurons (*Fagan et al., 2018*; *Macosko et al., 2009*; *Srinivasan et al., 2008*; *Wan et al., 2019*; *White et al., 2007*). Male *C. elegans* exhibit faster movement, potentially due to mechanisms at the interneuron level or neuromuscular junction (*Hao et al., 2023*; *Mowrey et al., 2014*; *Zeng et al., 2023*). Cook et al. reconstructed the neural networks of male and hermaphrodite *C. elegans* by compiling serial images acquired through electron microscopy. They identified 302 neurons in hermaphrodite and 385 neurons in male, along with their synaptic connections, including chemical and electrical synapses (*Cook et al., 2019*). These two networks serve as the foundation for understanding behavioral dimorphism at the whole brain level. In this study, we generated the functional neural networks of male and hermaphrodite *C. elegans* by merging neurons with the same function. Using graph theory and computational neuroscience methods, we systematically analyzed the structural and dynamical differences between these two neural networks. We identified circuitry mechanisms underlying the role of sex-specific neurons in the male neural network and predicted sexually dimorphic behaviors based on these structural and dynamical differences. Our predictions were validated by discovering a circuit with sex-shared neurons and dimorphic connections

responsible for enhanced local search behavior in males. This study provides evidence for the significant role of neural network architecture and the circuitry bases of dimorphic behaviors.

## Results

### Structural characteristics of functional neural networks in male and hermaphrodite *C. elegans*

We employed network science methods to systematically analyze the structural features of male and hermaphrodite neural networks (*Cook et al., 2019*). In order to simplify network size to emphasize the functional connectivity, we merged neurons with similar functions into a singular node. For instance, neurons ADAL (ADA Left) and ADAR (ADA Right) were consolidated as node ADA, and their total synaptic connections were integrated as the connections of ADA. Neurons with diverse functions were maintained as separate nodes, such as ASEL and ASER. We generated two simplified functional neural networks: the male neural network that comprises 161 neuron nodes and the hermaphrodite neural network that is composed of 122 nodes (*Figure 1—figure supplement 1*). In *Figure 1—figure supplement 1*, synaptic connections between neurons are denoted by clockwise lines. The line width corresponds to the synapse number, which is the sum of chemical synapses and electrical synapses (*Cook et al., 2019*). Given that chemical synapses are one-directional while electrical synapses are two-directional, we considered that one electrical synapse between two neurons is equivalent to two reciprocal chemical synapses to generate the connection matrix (*Supplementary file 1*). We found that 120 nodes are shared in these two neural networks (*Figure 1—figure supplement 2*), suggesting the dimorphic behaviors are largely mediated by male-specific neurons or synaptic connection differences among sex-shared neurons.

Given the directional nature and variable synapse numbers of the synaptic connections, we abstracted these neural networks as two directed-weighted graphs (*Figure 1—figure supplement 1*). We used graph theory methods designed for directed-weighted graphs to facilitate feature detection in both the male and hermaphrodite neural networks. One obvious observation is that the absence of connections between pharyngeal neurons and other neurons in the male neural network suggests a relatively independent functionality of pharyngeal neurons in males, though several studies imply the non-synaptic communications in *C. elegans* (*Dag et al., 2023*; *Shen et al., 2016*).

The node strength measures how strongly a node directly possesses with other nodes in the network. We calculated the node strength for each node and revealed that the distribution of node strengths in both hermaphrodite and male networks follows an exponential distribution (*Figure 1A*). Overall, neurons in the male network exhibit a larger node strength (*Figure 1B*). Comparing the node strengths of sex-specific neurons and sex-shared neurons, we found that the increased node strengths of sex-specific neurons in the male neural network, but not in the hermaphrodite network (*Figure 1C and D*), indicated the stronger connections of sex-specific neurons in the male neural network. A substantial node strength indicates a neuron's potential to exert dominance in shaping behavioral outputs. By classifying behavioral outputs of the top 20 neurons based on node strength, we predict enhanced mating and suppressed locomotion and chemotaxis behaviors in males (*Figure 1E*). Several sex-shared neurons are in the top 20 list in both neural networks, including AVA, ALA, and RIA, highlighting their important roles in both sexes (*Supplementary file 2*).

We further analyzed the node strength by dividing it into node out-strength and in-strength, representing the synaptic outputs or inputs of individual neurons, respectively. Both out-strength and in-strength exhibit an exponential distribution (*Figure 1F and J*). It is observed that neurons in males have stronger out-strength compared to those in hermaphrodites, particularly for male-specific neurons (*Figure 1G–I*). Interestingly, although the overall in-strength in both male and hermaphrodite neural networks is similar (*Figure 1K*), the analysis shows a difference in how sex-specific and sex-shared neurons receive inputs. Male-specific neurons receive more substantial input signals compared to sex-shared neurons in males (*Figure 1L*), suggesting a heightened sensitivity or importance in processing inputs related to sex-specific behaviors in males. In contrast, female-specific neurons in hermaphrodites exhibit the pattern of decreased in-strength (*Figure 1M*), highlighting a sex-dependent variation in neural network architecture and function.

The node strength takes into account the total synapse number of individual neurons, regardless of the number of connections with other neurons. Next, we calculated edge weight that is defined

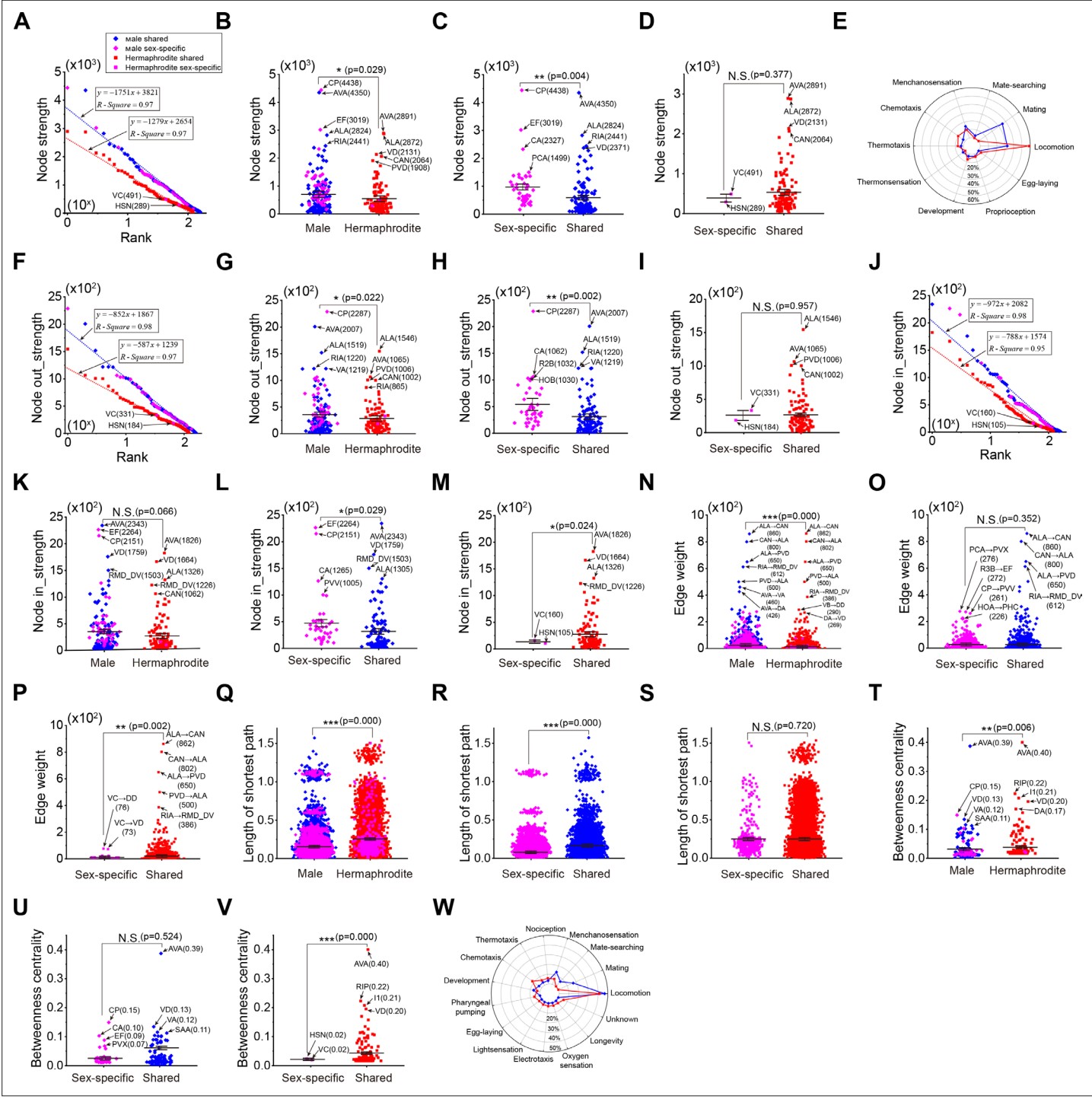

**Figure 1.** Structural characteristics of neural networks in male and hermaphrodite. (**A**) The node strengths follow exponential distribution in both neural networks. (**B**) Neurons in the male neural network exhibit significantly more node strengths in comparison with those in the hermaphrodite. (**C**) Sex-specific neurons exhibit significantly more node strengths in comparison with sex-shared neurons in the male neural network. (**D**) Sex-specific neurons exhibit similar node strengths in comparison with sex-shared neurons in the hermaphrodite neural network. (**E**) Prediction of behavioral outputs based on the top 20 node strengths in male and hermaphrodite neural networks. (**F**) The node out-strengths follow exponential distribution in both neural networks. (**G**) Neurons in the male neural network exhibit significantly more node out-strengths in comparison with those in the hermaphrodite. (**H**) Sex-specific neurons exhibit significantly more node out-strengths in comparison with sex-shared neurons in the male neural network. (**I**) Sex-specific neurons exhibit similar node out-strengths in comparison with sex-shared neurons in the hermaphrodite neural network. (**J**) The node in-strengths follow exponential distribution in both neural networks. (**K**) Neurons in the male neural network exhibit slightly more node in-strengths in comparison with those in the hermaphrodite. (**L**) Sex-specific neurons exhibit significantly more node in-strengths in comparison with sex-shared neurons in the

*Figure 1 continued*

male neural network. (**M**) Sex-specific neurons exhibit significantly less node in-strengths in comparison with sex-shared neurons in the hermaphrodite neural network. (**N**) Neurons in the male neural network exhibit significantly more edge weights in comparison with those in the hermaphrodite. (**O**) Sex-specific neurons exhibit similar edge weights in comparison with sex-shared neurons in the male neural network. (**P**) Sex-specific neurons exhibit significantly less edge weights in comparison with sex-shared neurons in the hermaphrodite neural network. (**Q**) Neurons in the male neural network exhibit significantly shorter length of the shortest path in comparison with those in the hermaphrodite. (**R**) Sex-specific neurons exhibit significantly shorter length of the shortest path in comparison with sex-shared neurons in the male neural network. (**S**) Sex-specific neurons exhibit similar length of the shortest path in comparison with sex-shared neurons in the hermaphrodite neural network. (**T**) Neurons in the male neural network exhibit significantly lower betweenness centrality in comparison with those in the hermaphrodite. (**U**) Sex-specific neurons exhibit significantly lower betweenness centrality in comparison with sex-shared neurons in the male neural network. (**V**) Sex-specific neurons exhibit significantly lower betweenness centrality in comparison with sex-shared neurons in the hermaphrodite neural network. (**W**) Prediction of behavioral outputs based on the top 20 betweenness centrality in male and hermaphrodite neural networks. Student's *t*-test for all the statistical tests. *$p<0.05$, **$p<0.01$, ***$p\le0.001$.

The online version of this article includes the following figure supplement(s) for figure 1:

**Figure supplement 1.** Directed-weighted graphs of male and hermaphrodite neural networks.

**Figure supplement 2.** The majority of neurons are shared in male and hermaphrodite neural networks.

**Figure supplement 3.** Weight comparison for the top edges in male neural network, including ALA, AVA, and RIA-related edges.

**Figure supplement 4.** Predicted circuits for sensorimotor integration in males (**A**) and hermaphrodites (**B**).

**Figure supplement 5.** An artificial network as an example to demonstrate the calculation of each graph theory parameter.

by assigning a number to each edge, representing the synapse number of the individual connection between two neurons. Our analysis indicated that, on average, male neural networks tend to have larger edge weights compared to hermaphrodite networks (*Figure 1N*). This suggests that the male network may have stronger synaptic connections per connection on average. A closer look at the edge weights of sex-specific and sex-shared neuron-related edges reveals a nuanced picture. In males, the edge weights associated with sex-specific neurons are comparable to those associated with sex-shared neurons (*Figure 1O*). This indicates a uniformity in how connections are established within the male neural network, regardless of whether the neurons are involved in sex-specific or shared functions. However, in hermaphrodites, the edge weights of sex-specific neurons are significantly smaller than those of sex-shared neurons (*Figure 1P*), suggesting that synaptic connections in the hermaphrodite network may be organized differently, with decreased modification of sexual-related behaviors.

Focusing on the top edge weights in the male neural network, we found that the consistent high weights of ALA-related edges in both sexes underscore the importance of ALA-mediated harsh touch escape behavior for individuals of both males and hermaphrodites (*Figure 1N*, *Figure 1—figure supplement 3*). However, in males, the weights of edges associated with AVA and RIA are notably higher (*Figure 1N*, *Figure 1—figure supplement 3*), indicating the increased AVA- or RIA-mediated behaviors in males.

In graph theory, the shortest path problem involves finding a path between two nodes in a graph that minimizes the sum of the weights. For a neural network represented as a graph, the shortest path between two neurons can provide insight into the functional correlation between them, especially when they are not directly connected. A shorter shortest path in the male neural network suggests a more cohesive interconnection of functional relationships among neurons in males (*Figure 1Q*). When comparing the shortest paths between two sex-shared neurons or those involving at least one sex-specific neuron, it is observed that the latter shows a tighter connection in males compared to hermaphrodites (*Figure 1R and S*).

Betweenness centrality in a graph measures the number of shortest paths that pass through a particular node, thus highlighting the key functional nodes within the network. The top betweenness centrality of AVA in both male and hermaphrodite networks (*Figure 1T–V*) underscores the critical role of AVA in both sexes. By examining the top 20 neurons ranked by betweenness centrality, we infer that males exhibit enhanced mechanosensation, mate-searching, and mating behaviors (*Figure 1W*). These predictions are based on the understanding that nodes with high betweenness centrality are often involved in a significant number of shortest paths, indicating that they are integral to the network's function and connectivity. AVA, in particular, may be central to various behaviors and processes, including mechanosensory responses and social interactions such as mate-searching and mating.

Acknowledging the crucial influence of sex-specific neurons and their interactions in driving behaviors unique to each sex (*Fenk and de Bono, 2015*; *Hart and Hobert, 2015*; *Huang et al., 2023*; *Koo et al., 2011*; *Susoy et al., 2021*), we generated the sex-specific sub-neural networks that include the sex-specific neurons along with their directed-connected sex-shared neurons. This approach allows us to delve into the intrinsic properties of neural networks that exhibit sexual dimorphism (*Figure 2—figure supplement 1*).

Our analysis of sex-specific sub-neural networks has revealed that they share similar characteristics with their respective entire neural networks. Node strengths, which include total node strength, node out-strength, and node in-strength, are found to follow exponential distributions within these sex-specific sub-networks for both males and hermaphrodites (*Figure 2A, E and I*). In males, the sex-specific sub-neural network exhibits enhanced node strengths when compared to the hermaphrodite's, indicating a more robust connectivity within the male network (*Figure 2B*). And the male sex-specific neurons reveal larger total node strengths and node out-strengths (*Figure 2C and G*), while the node in-strengths remain similar to those of hermaphrodites (*Figure 2K*). In contrast, hermaphrodites show no significant difference in total strengths, out-strengths, and in-strengths between sex-specific neurons and sex-shared neurons (*Figure 2D, H and L*). The large edge weights and short length of the shortest path within the male sex-specific sub-neural network suggest a tighter functional connection, indicative of a more integrated network (*Figure 2M–R*). Specifically, the top edge weights of AIY→RIA and RIA→RMD_DV in the male sub-neural network, along with the role of RIA in sensorimotor integration for olfactory steering, suggest that this AIY→RIA→RMD_DV circuit plays a significant role in mediating chemotaxis-related behaviors in males (*Figure 2M*). Additionally, the high betweenness centrality of AVA and the top edge weights of AVA→DA and AVA→VA within the male sub-neural network point to AVA's regulatory role in locomotion, which is essential for male's sex-related behaviors (*Figure 2M and S*). These findings underscore the importance of understanding the connectivity and functional roles of neurons within sex-specific neural networks and how these factors contribute to the manifestation of sex-specific behaviors.

Previous studies have demonstrated that sex-shared neurons can form dimorphic circuits that are responsible for mediating behaviors that vary between males and hermaphrodites (*Serrano-Saiz et al., 2017*; *Setty et al., 2022*; *White et al., 2007*). Therefore, we generated the sex-shared sub-neural networks containing sex-shared neurons and the connections among them (*Figure 3—figure supplement 1*) and analyzed the characteristics of these sex-shared sub-neural networks for both sexes.

Interestingly, our findings indicate that sex-shared sub-neural networks display distinct features when compared to the complete neural networks. Both of these sub-networks exhibit node strengths that follow a similar exponential distribution (*Figure 3A, B, D, E, G and H*). To quantify the differences in node strength between two sexes, we calculated the node strength difference for each neuron, defined as the node strength of the male neuron minus that of the hermaphrodite neuron (*Figure 3C, F and I*). Significantly, some neurons show an increase in node strength in males, including the commander neuron AVA, motor neurons VA and VB, and the interneuron RIA (*Figure 3C, F and I*). This increase in node strength suggests that the neural functions associated with AVA and RIA may be heightened in males. Conversely, the comparatively reduced node strength observed in several neurons in males could be indicative of behaviors that are specific to hermaphrodites, such as those related to neurons URY, PVC, and PVR (*Figure 3C, F and I*).

In examining the edge weight feature of the neural networks, we identified several edges that are specific to one sex (*Supplementary file 3*). These findings underscore the potential influence of these sex-specific edges on the generation of dimorphic behavioral outputs. For clarity, we focus on the top five weighted sex-specific edges and their edge counts, as indicated in *Figure 3J*. When comparing the weights of shared edges, it is observed that they tend to be more substantial in males (*Figure 3K*). This disparity in edge weight is particularly notable in circuits related to RIA, such as AIY→RIA→RMD_DV, and circuits associated with AVA, including AVA→DA and AVA→VA, which are markedly enriched in males. On the other hand, circuits like PVP-AQR and those involving FLP neurons, such as FLP→AVD and FLP→AVB, appear to be more dominant in hermaphrodites (*Figure 3L*). These circuit-specific differences suggest that they may contribute to distinct behavioral manifestations between two sexes.

Furthermore, we observed a decrease in the shortest paths among pharyngeal neurons, such as M3, M5, and NSM, in the male neural network (*Figure 3M and N*). Along with a lack of connections

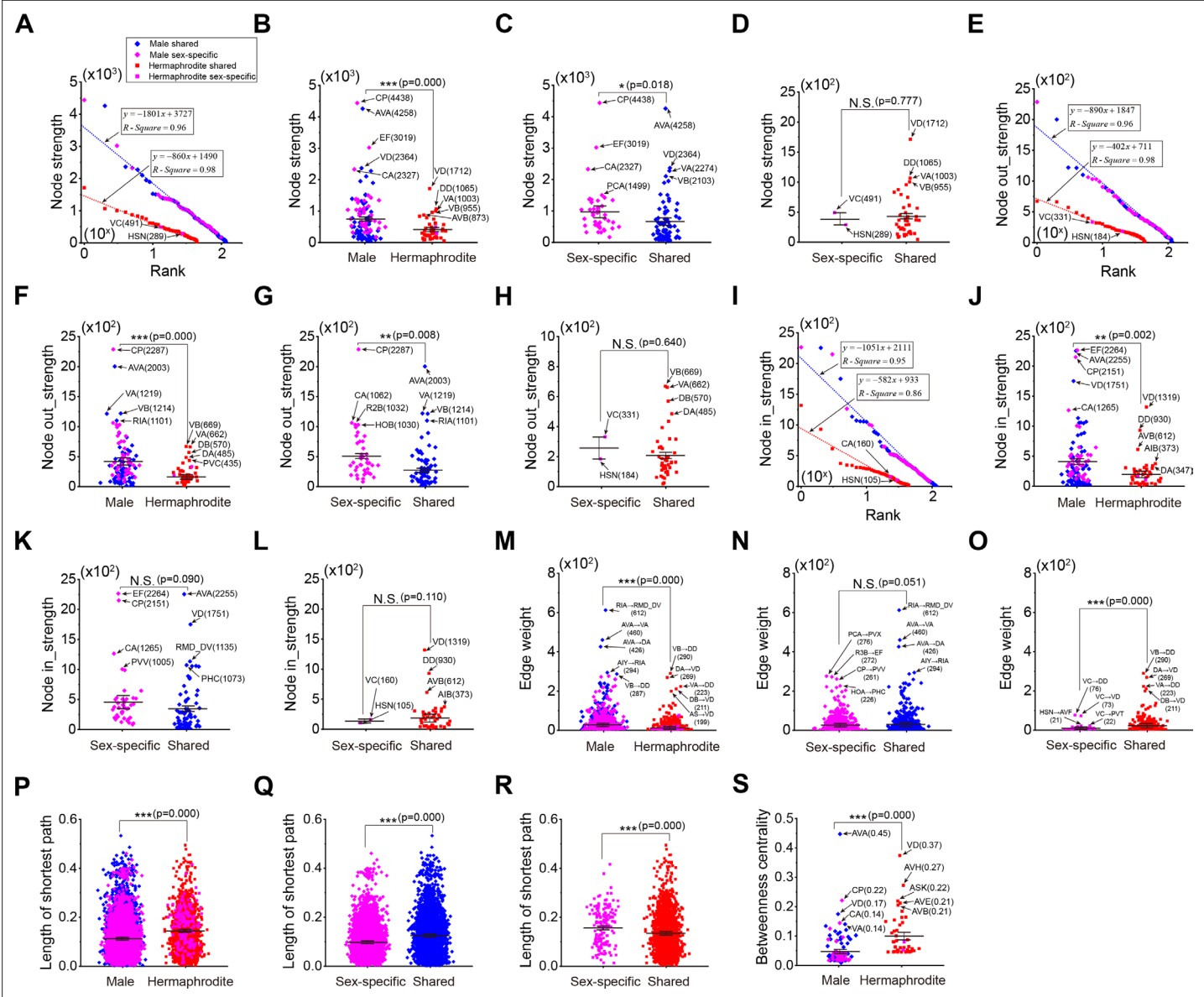

**Figure 2.** Structural characteristics of sex-specific sub-neural networks in male and hermaphrodite. (**A**) The node strengths follow exponential distribution in both sex-specific sub-neural networks. (**B**) Neurons in the male sex-specific sub-neural network exhibit significantly more node strengths in comparison with those in the hermaphrodite. (**C**) Sex-specific neurons exhibit significantly more node strengths in comparison with sex-shared neurons in the male sex-specific sub-neural network. (**D**) Sex-specific neurons exhibit similar node strengths in comparison with sex-shared neurons in the hermaphrodite sex-specific sub-neural network. (**E**) The node out-strengths follow exponential distribution in both sex-specific sub-neural networks. (**F**) Neurons in the male sex-specific sub-neural network exhibit significantly more node out-strengths in comparison with those in the hermaphrodite. (**G**) Sex-specific neurons exhibit significantly more node out-strengths in comparison with sex-shared neurons in the male sex-specific sub-neural network. (**H**) Sex-specific neurons exhibit similar node out-strengths in comparison with sex-shared neurons in the hermaphrodite sex-specific sub-neural network. (**I**) The node in-strengths follow exponential distribution in both sex-specific sub-neural networks. (**J**) Neurons in the male sex-specific sub-neural network exhibit significantly more node in-strengths in comparison with those in the hermaphrodite. (**K**) Sex-specific neurons exhibit slightly more node in-strengths in comparison with sex-shared neurons in the male sex-specific sub-neural network. (**L**) Sex-specific neurons exhibit similar node in-strengths in comparison with sex-shared neurons in the hermaphrodite sex-specific sub-neural network. (**M**) Neurons in the male sex-specific sub-neural network exhibit significantly more edge weights in comparison with those in the hermaphrodite. (**N**) Sex-specific neurons exhibit slightly less edge weights in comparison with sex-shared neurons in the male sex-specific sub-neural network. (**O**) Sex-specific neurons exhibit significantly less edge weights in comparison with sex-shared neurons in the hermaphrodite sex-specific sub-neural network. (**P**) Neurons in the male sex-specific sub-neural network exhibit significantly shorter length of the shortest path in comparison with those in the hermaphrodite. (**Q**) Sex-specific neurons exhibit significantly shorter length of the shortest path in comparison with sex-shared neurons in the male sex-specific sub-neural network. (**R**) Sex-specific neurons exhibit significantly longer length of the shortest path in comparison with sex-shared neurons in the hermaphrodite sex-specific sub-neural

*Figure 2 continued*

network. (**S**) Neurons in the male sex-specific sub-neural network exhibit significantly lower betweenness centrality in comparison with those in the hermaphrodite. Student's *t*-test for all the statistical tests. *p<0.05, **p<0.01, ***p≤0.001.

The online version of this article includes the following figure supplement(s) for figure 2:

**Figure supplement 1.** Sex-specific sub-neural networks of hermaphrodite (**A**) and male (**B**).

**Figure supplement 2.** Predicted male-specific (**A**) and hermaphrodite-specific (**B**) circuits.

with other neurons (*Figure 1—figure supplement 1*), this observation indicates the formation of a functional unit consisting of pharyngeal neurons, which may be critical for male-specific behaviors. We have noted an increase in the shortest paths involving the RIP neuron, such as RIP→ASI, RIP→ADE, RIP→ALN, RIP→ASJ, and RIP→ASER (*Figure 3N*), along with a decrease in the betweenness centrality of the RIP neuron (*Figure 3O and P*). These changes strongly suggest an enhanced dimorphic function of the RIP neuron in hermaphrodites, which could be integral to hermaphrodite-specific behaviors.

## Dynamical characteristics of functional neural networks in male and hermaphrodite *C. elegans*

To systemically study the dynamics of these two neural networks, we performed computational neuroscience methods to assess the functional role of individual neurons in the network. We utilized a non-spiking RC model to simulate neuronal activity because it is widely accepted that neurons in *C. elegans* are non-spiked, with the acknowledgment of action potentials in several neurons (*Jiang et al., 2022*; *Liu et al., 2018b*). To simplify the simulation, we standardized parameters for all neurons, maintaining a resting potential of –60 mV and the fixed linear correlation of the current input and synapse number (*Source code 1*). We stimulated individual neurons by shifting the membrane potential to +60 mV for a duration of 40 seconds and monitored the membrane potential changes of all the other neurons. *Figure 4A* illustrates the stimulation of AVA in hermaphrodite neural network as an example; the other neurons of hermaphrodite neural network exhibit diverse response patterns (*Figure 4A*). From these simulations, we identified the neuron that exhibited the strongest response when an individual neuron was stimulated, creating functional pairs. We found that only 30% of these functional pairs are conserved across both neural networks (*Figure 4B* and *Supplementary file 4*). To gauge the influence of a neuron as an upstream component within the network, we measured the cumulative changes in membrane potential across all other neurons when a particular neuron was stimulated. A substantial alteration in membrane potential across all other neurons would indicate that a particular neuron has a significant impact on the entire network as an upstream component. According to our results, neurons in the male network are more capable of activating the network than those in the hermaphrodite network (*Figure 4C*). Furthermore, sex-specific neurons, when acting as upstream neurons, have a greater effect on the network in males than do sex-shared neurons (*Figure 4D*), a trend not observed in hermaphrodites (*Figure 4E*). Similar patterns were observed when assessing the dynamic functions of neurons as part of the downstream component of the network (*Figure 4G–I*). By categorizing the top 20 upstream and top 20 downstream neurons, we attempted to predict sexually dimorphic behaviors. Both predictions highlighted mating behavior as a key aspect of male-specific behaviors (*Figure 4F and J*).

## A neural circuit for dimorphic spontaneous local search behavior

Given the structural and dynamical discrepancies identified between the male and hermaphrodite neural networks, several sexually dimorphic behaviors can be predicted. In particular, male behavior is primarily characterized by an emphasis on sexual-related behaviors. The AVA neuron emerges as pivotal to the function of the male network, participating in both sex-specific and sex-shared sub-neural networks. Considering that AVA regulates local search behavior in hermaphrodite (*Gray et al., 2005*; *López-Cruz et al., 2019*) and enhanced connections of AVA in the male neural network, therefore, we examined the hypothesis that AVA-related neural circuit could be a key regulatory component for dimorphic spontaneous local search behavior in males.

We placed individual worms onto distinct food patches and recorded the number of reversals over the initial ten minutes in the new environment; this number indicates local search behavior (*Gray et al., 2005*). Results showed that males display high levels of local search behavior on both thick

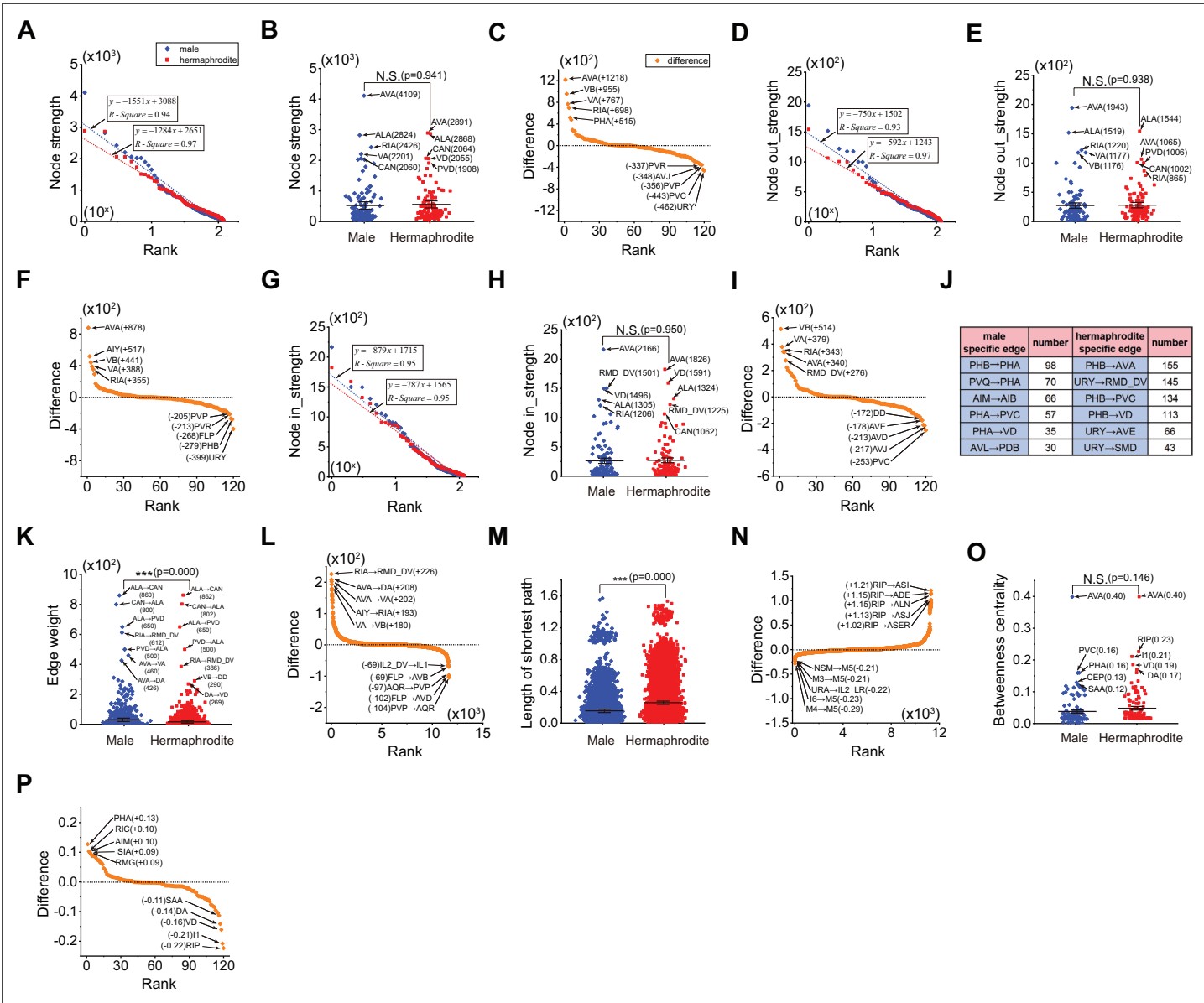

**Figure 3.** Structural characteristics of sex-shared sub-neural networks in male and hermaphrodite. (**A**) The node strengths follow exponential distribution in both sex-shared sub-neural networks. (**B**) Neurons in the male sex-shared sub-neural network exhibit similar node strengths in comparison with those in the hermaphrodite. (**C**) Node strength differences of the same neurons in two sex-shared sub-neural networks, positive represents more strength in male. (**D**) The node out-strengths follow exponential distribution in both sex-shared sub-neural networks. (**E**) Neurons in the male sex-shared sub-neural network exhibit similar node out-strengths in comparison with those in the hermaphrodite. (**F**) Node out-strength differences of the same neurons in two sex-shared sub-neural networks, positive represents more strength in male. (**G**) The node in-strengths follow exponential distribution in both sex-shared sub-neural networks. (**H**) Neurons in the male sex-shared sub-neural network exhibit similar node in-strengths in comparison with those in the hermaphrodite. (**I**) Node in-strength differences of the same neurons in two sex-shared sub-neural networks, positive represents more strength in male. (**J**) Top 5 male-specific and hermaphrodite-specific edges. (**K**) Neurons in the male sex-shared sub-neural network exhibit significantly more edge weights in comparison with those in the hermaphrodite. (**L**) Weight differences of the connection between same neurons in two sex-shared sub-neural networks, positive represents more strength in male. (**M**) Neurons in the male sex-shared sub-neural network exhibit significantly shorter length of the shortest path in comparison with those in the hermaphrodite. (**N**) Length differences of the shortest length between same neurons in two sex-shared sub-neural networks, positive represents longer length in male. (**O**) Neurons in the male sex-shared sub-neural network exhibit significantly lower betweenness centrality in comparison with those in the hermaphrodite. (**P**) Betweenness centrality differences of the same neurons in two sex-shared sub-neural networks, positive represents larger betweenness centrality in male. Student's *t*-test for all the statistical tests. *p<0.05, **p<0.01, ***p≤0.001.

The online version of this article includes the following figure supplement(s) for figure 3:

**Figure supplement 1.** Sex-shared sub-neural networks of hermaphrodite (**A**) and male (**B**).

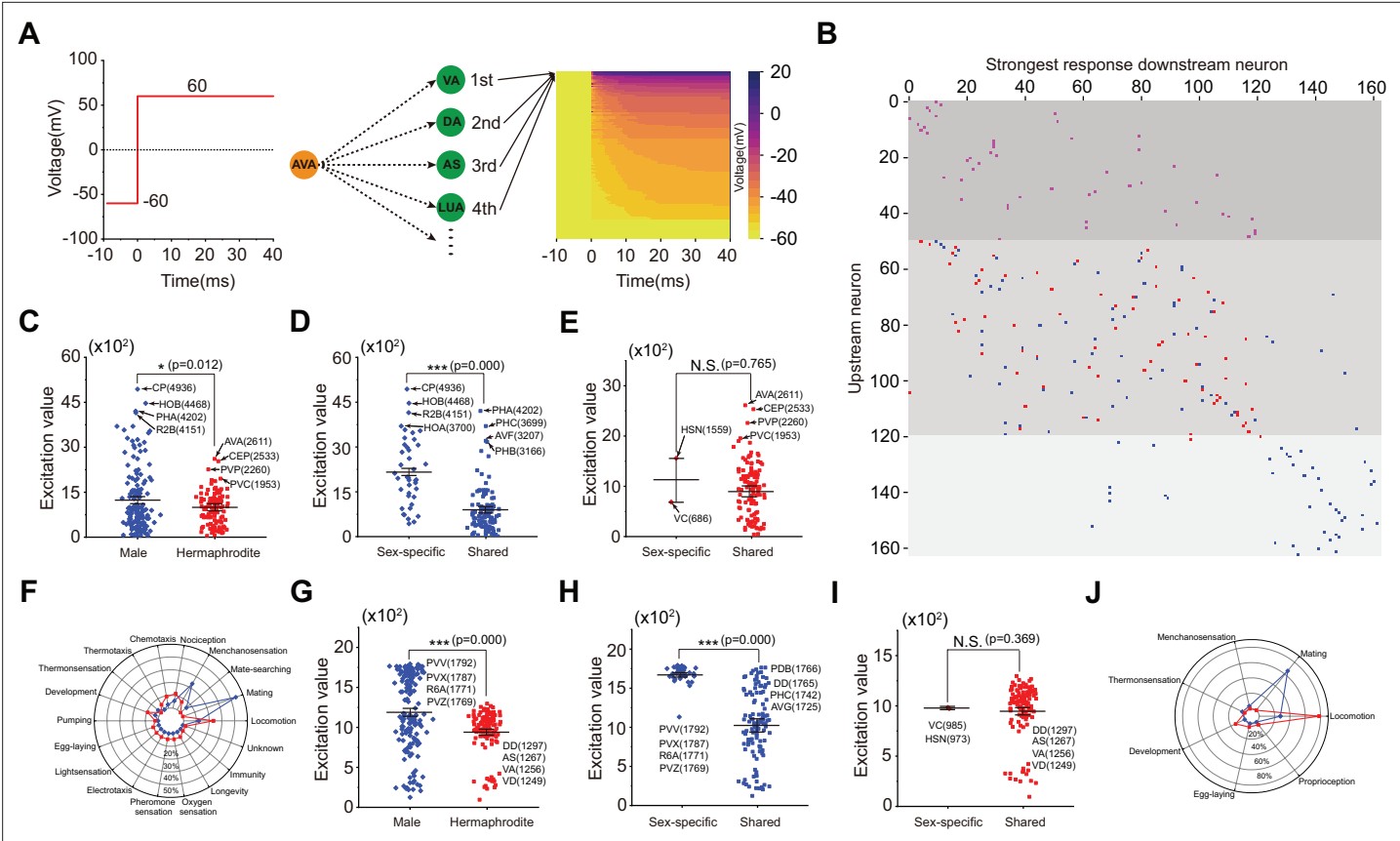

**Figure 4.** Dynamical characteristics of neural networks in male and hermaphrodite. (**A**) Showcase of AVA neuron in hermaphrodite, left represents protocol of manipulating AVA membrane potential, right represents diverse membrane potential dynamics of all the other neurons in hermaphrodite neural network. (**B**) Downstream neurons with the strongest responses, pink color represents the same strongest response downstream neurons in two neural networks. Blue color represents strongest response downstream neurons in male, red color represents strongest response downstream neurons in hermaphrodite. (**C**) Sum of membrane potential changes in all the other neurons accesses individual neuron's role as upstream neuron in a neural network. Neurons in the male neural network exhibit stronger influence as upstream neuron in comparison with those in the hermaphrodite. (**D**) Sex-specific neurons exhibit significantly higher membrane potential changes in all the other neurons in comparison with sex-shared neurons in the male neural network. (**E**) Sex-specific neurons exhibit similar membrane potential changes in all the other neurons in comparison with sex-shared neurons in the hermaphrodite neural network. (**F**) Prediction of behavioral outputs based on the top 20 neurons as upstream in male and hermaphrodite neural networks. (**G**) Sum of membrane potential changes in a neuron when activating all the other neurons accesses individual neuron's role as downstream neuron in a neural network. Neurons in the male neural network exhibit stronger influence as downstream neuron in comparison with those in the hermaphrodite. (**H**) Sex-specific neurons exhibit significantly higher membrane potential changes when activating all the other neurons in comparison with sex-shared neurons in the male neural network. (**I**) Sex-specific neurons exhibit similar membrane potential changes when activating all the other neurons in comparison with sex-shared neurons in the hermaphrodite neural network. (**J**) Prediction of behavioral outputs based on the top 20 neurons as downstream in male and hermaphrodite neural networks. Student's *t*-test for all the statistical tests. *p<0.05, **p<0.01, ***p≤0.001.

and thin food patches, as well as patches without food (*Figure 5A-D*). The absence of food alters the initial local search behavior in hermaphrodites, with no impact on males (*Figure 5C and D*). The males' enhanced local search behavior is dependent on sexual maturity, as evidenced by the similar reversal rates of late larval males and hermaphrodites (*Figure 5E*). To explore the potential impact of the prior experiences during development on male's local search behavior, we separated L4 males on three distinct types of plates: a plate devoid of both food and hermaphrodites, a plate containing food without hermaphrodites or a plate with both food and hermaphrodites. There was no significant difference among males with different prior experiences, suggesting that developmental experiences do not influence the enhanced local search behavior in males (*Figure 5F*). Overall, we discovered a sexually dimorphic behavior in *C. elegans*, that sexually matured males exhibit enhanced local search behavior regardless of the environment.

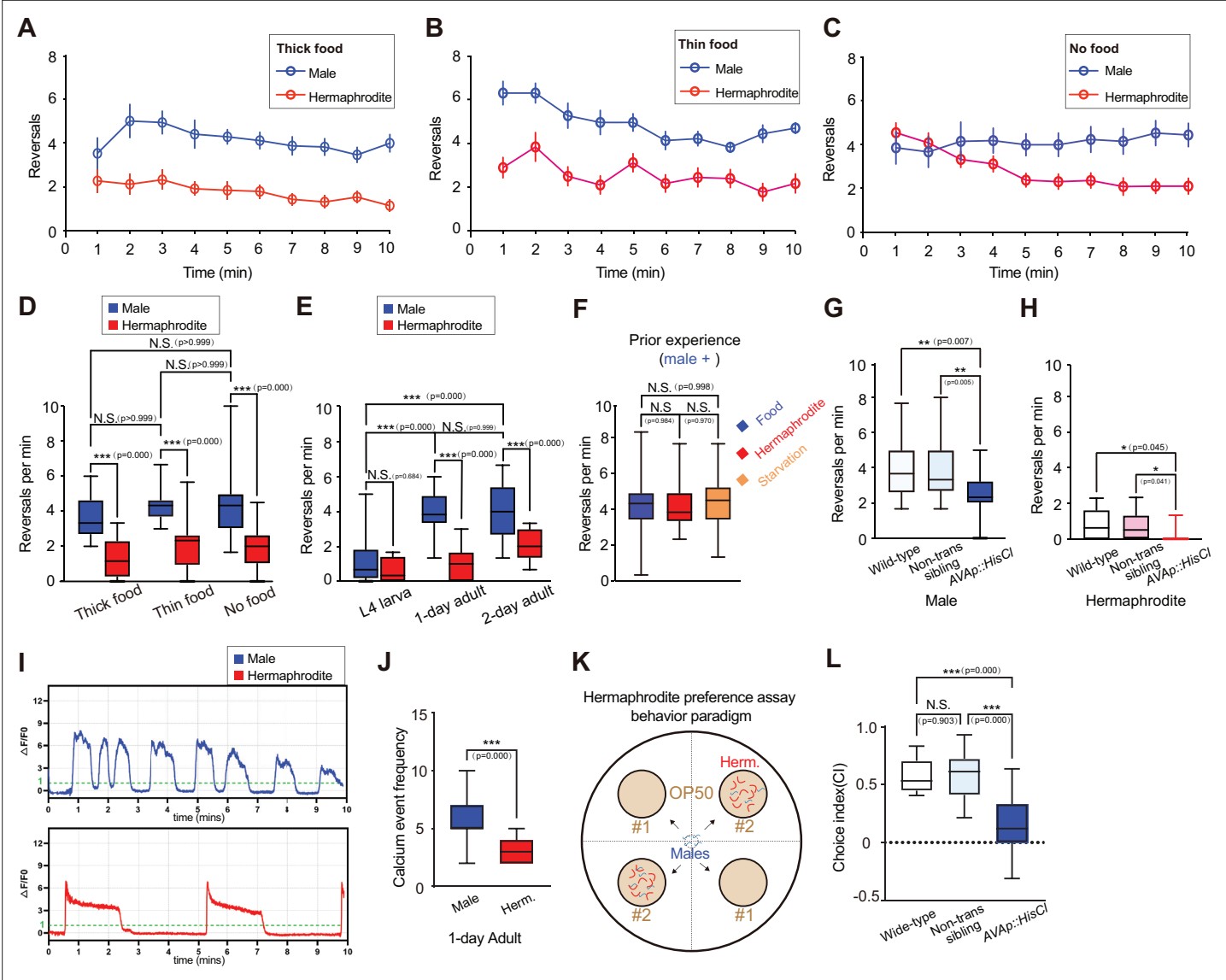

**Figure 5.** Males reveal consistently high local search behavior. Reversal rate within a 10-minute time window for N2 wild-type males and hermaphrodites on 3.5 cm OP50-seeded plates under three conditions: (**A**) Thick food (10 μL *E. coli* OP50 cultured for 16 hours). (**B**) Thin food (10 μL *E. coli* OP50 cultured for 0.5 hours). (**C**) No food. (**D**) Statistical testing for the average reversal rate between minutes 7–10 under Thick food, Thin food, and No food conditions, with sample sizes of 17, 23, and 21 respectively. (**E**) Statistical testing for the average reversal rate between minutes 7–10 under different developmental stages, Sample size: 17 males and 14 hermaphrodites in L4 stage; 20 males and 25 hermaphrodites in 1-day adult stage; 2-day adult: 16 males and 15 hermaphrodites in 2-day adult stage. (**F**) Statistical testing for the average reversal rate between minutes 7–10 with different social experiences, with sample sizes of 20 for each group. (**G**) Statistical testing for the average reversal rate between minutes 7–10 of 1-day adult males under different genetic backgrounds. 25 wild-type worms, 25 AVA::HisCl1 transgenic worms, and 21 non-transgenic siblings. (**H**) Statistical testing for the average reversal rate between minutes 7–10 of 1-day adult hermaphrodites under different genetic backgrounds. 15 Wild-type worms, 15 AVA::HisCl1 transgenic worms, and 18 non-transgenic siblings. (**I**) Showcases of spontaneous calcium events in AVA of male and hermaphrodite 1-day adults. (**J**) Statistical testing for the spontaneous calcium event frequency in AVA of male and hermaphrodite 1-day adults. Sample size: 16 males and 13 hermaphrodites. (**K**) Hermaphrodite preference assay behavior paradigm. (**L**) Statistical testing for the choice index of 1-day adult males under different genetic backgrounds, with sample sizes of 11 for each group. Student's *t*-test for all the statistical tests. *p<0.05, **p<0.01, ***p≤0.001.

The online version of this article includes the following figure supplement(s) for figure 5:

**Figure supplement 1.** Similar calcium event frequency of AVA in males (**A**) and hermaphrodites (**B**) L4 stage.

To examine AVA's role in local search behavior, we temporally suppressed AVA activity by expressing HisCl1 (histamine-gated chloride channel) in AVA and administrating histamine (*Pokala et al., 2014*). The significantly reduced reversal rates indicate the commanding role of AVA in driving reversals in both males and hermaphrodites (*Figure 5G and H*). Subsequently, we conducted in vivo calcium imaging studies on AVA neurons. Our findings revealed that AVA neurons exhibit spontaneous calcium transients, with adult males showing a significantly higher frequency of these events compared to hermaphrodites (*Figure 5I and J*). However, this difference is not observed in late larval males (*Figure 5—figure supplement 1*), which aligns with the behavioral outcomes.

Next, to identify male's local search target, we assessed male's behavioral preference towards hermaphrodites. We generated four food patches with equal distance to the center of the plate, then placed approximately 30 adult hermaphrodites on diagonal food patches for 1 hour, followed by introducing approximately 30 males to the center of the plate (*Figure 5K*). After 1 hour, we quantified the distribution of males across food patches with and without hermaphrodites. A greater proportion of wild-type males congregated on patches containing hermaphrodites. However, inhibiting AVA activity notably reduced the preference for hermaphrodites (*Figure 5L*). These findings suggest that AVA-mediated enhancement of male local search behavior aids in locating hermaphrodites for mating purposes.

In order to elucidate the AVA-related dimorphic circuit responsible for male-specific local search behavior, we quantified the differences in chemical synaptic connections with AVA neurons between males and hermaphrodites. Notably, PQR, RIC, and DVC neurons demonstrated the highest synaptic output to AVA (*Figure 6A and B*, *Figure 6—figure supplement 1*), suggesting their potential role as upstream neurons. Consequently, we investigated the effect of PQR, RIC, and DVC on male's enhanced local search behavior by blocking their synaptic release using TeTx expression driven by cell-specific promoters (*Figure 6—figure supplement 2*). Our findings indicated that selectively blocking either RIC or DVC significantly reduced the local search behavior in males (*Figure 6C and E*), but not in hermaphrodites (*Figure 6D and F*). However, PQR is not involved in local search regulation in both male and hermaphrodite (*Figure 6G, H* and *Figure 6—figure supplement 3*). To further validate these results, we conducted AVA calcium imaging under conditions where PQR, DVC, or RIC synaptic activity was blocked. A decrease in the frequency of spontaneous calcium transients was observed specifically when DVC or RIC activity was suppressed, indicating that both DVC and RIC are indeed upstream neurons of AVA and play a crucial role in modulating the enhanced local search behavior observed in males (*Figure 6I-N*). These results are consistent with the behavioral observations.

DVC identified as a glutamatergic neuron (*Pereira et al., 2015*) and activates AVA calcium transient events for local search behavior in males (*Figure 6E and K*). Therefore, we tested whether NMDA or AMPA receptors on AVA regulate local search behavior. We generated transgenic strains that individually silence NMDA and AMPA receptor genes in AVA neurons (*Figure 7A*). By examining the reversal rates, we evaluated the impact on local search behavior. Our results indicated that individually silencing AMPA receptor gene glr-2, as well as NMDA receptor genes, nmr-1 or nmr-2, led to a significant decrease in reversal rates in males (*Figure 7B, D and F*). However, no such effect was observed in hermaphrodites (*Figure 7C, E and G*). This suggests that both AMPA and NMDA receptors are crucial for mediating local search behavior in males. Furthermore, since RIC is known to release octopamine (*Pereira et al., 2015*), we explored the influence of octopamine receptors SER-3 and SER-6 in AVA on local search behavior (*Figure 7A*). By silencing the ser-3 and ser-6 genes specifically in AVA neurons and assessing the reversal rates, we aimed to determine their contribution. Our findings demonstrated that silencing ser-3 in AVA significantly reduced the reversal rate in males, without affecting hermaphrodites (*Figure 7H and I*). In contrast, silencing ser-6 in AVA had no discernible effect on the reversal rate in either sex (*Figure 7J and K*, *Figure 7—figure supplement 1*). These results suggest that the octopamine receptor SER-3 plays a specific role in regulating the increased local search behavior observed in males.

## Discussion

In this study, we systematically analyze the structural and dynamical differences of neural networks, which set the circuitry basis for the dimorphic behaviors in male and hermaphrodite *C. elegans*. We reveal that male neural network characteristics determine the prioritized sexual-related behaviors in males. We predict dimorphic behavioral outputs according to the structural and dynamical features

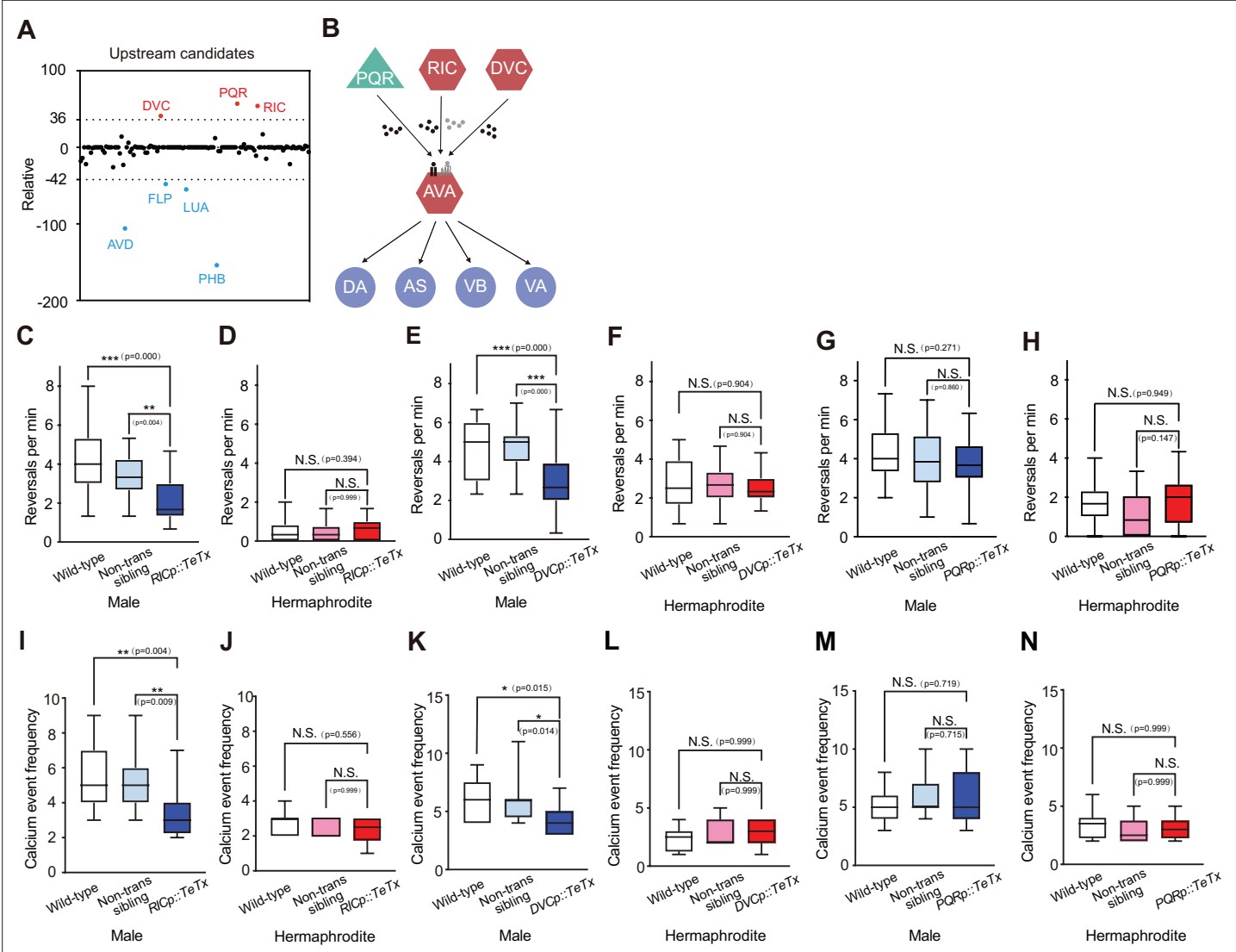

**Figure 6.** Neurons that are responsible for the increased local search in males. (**A**) Chemical synapse number difference of AVA with upstream neurons between male and hermaphrodite. (**B**) A circuit with AVA and top 3 upstream neurons and top 4 downstream neurons based on the chemical synapse number difference. (**C**) Statistical testing for the average reversal rate between minutes 7–10 of 1-day adult males under different genetic backgrounds for RIC neuron manipulation. 17 wild-type worms, 20 RIC::TeTx transgenic worms, and 20 non-transgenic siblings. (**D**) Statistical testing for the average reversal rate between minutes 7–10 of 1-day adult hermaphrodites under different genetic backgrounds for RIC neuron manipulation. 15 wild-type worms, 20 RIC::TeTx transgenic worms, and 15 non-transgenic siblings. (**E**) Statistical testing for the average reversal rate between minutes 7–10 of 1-day adult males under different genetic backgrounds for DVC neuron manipulation. 15 wild-type worms, 20 DVC::TeTx transgenic worms, and 15 non-transgenic siblings. (**F**) Statistical testing for the average reversal rate between minutes 7–10 of 1-day adult hermaphrodites under different genetic backgrounds for DVC neuron manipulation. 20 wild-type worms, 20 DVC::TeTx transgenic worms, and 18 non-transgenic siblings. (**G**) Statistical testing for the average reversal rate between minutes 7–10 of 1-day adult males under different genetic backgrounds for PQR neuron manipulation. 17 wild-type worms, 20 PQR::TeTx transgenic worms, and 20 non-transgenic siblings. (**H**) Statistical testing for the average reversal rate between minutes 7–10 of 1-day adult hermaphrodites under different genetic backgrounds for PQR neuron manipulation. 14 wild-type worms, 15 PQR::TeTx transgenic worms, and 14 non-transgenic siblings. (**I**) Statistical testing for the spontaneous calcium event frequency in AVA of 1-day adult males under different genetic backgrounds for RIC neuron manipulation. 11 wild-type worms, 20 RIC::TeTx transgenic worms, and 15 non-transgenic siblings. (**J**) Statistical testing for the spontaneous calcium event frequency in AVA of 1-day adult hermaphrodites under different genetic backgrounds for RIC neuron manipulation. 8 wild-type worms, 10 RIC::TeTx transgenic worms, and 10 non-transgenic siblings. (**K**) Statistical testing for the spontaneous calcium event frequency in AVA of 1-day adult males under different genetic backgrounds for DVC neuron manipulation. 10 wild-type worms, 13 DVC::TeTx transgenic worms, and 13 non-transgenic siblings. (**L**) Statistical testing for the spontaneous calcium event frequency in AVA of 1-day adult hermaphrodites under different genetic backgrounds for DVC neuron manipulation. 8 wild-type worms, 11 DVC::TeTx transgenic worms, and 11 non-transgenic siblings. (**M**) Statistical testing for the spontaneous calcium event frequency in AVA of 1-day adult males under different genetic backgrounds for PQR neuron manipulation. 11 wild-type worms, 14 PQR::TeTx transgenic worms, and 8 non-transgenic siblings. (**N**) Statistical testing for the spontaneous calcium event frequency

*Figure 6 continued on next page*

*Figure 6 continued*

in AVA of 1-day adult hermaphrodites under different genetic backgrounds for PQR neuron manipulation. Eight wild-type worms, eight PQR::TeTx transgenic worms, and eight non-transgenic siblings. Student's *t*-test for all the statistical tests. *p<0.05, **p<0.01, ***p≤0.001.

The online version of this article includes the following figure supplement(s) for figure 6:

**Figure supplement 1.** Chemical synapse number difference of AVA with downstream neurons between male and hermaphrodite.

**Figure supplement 2.** The expression of mCherry linked with TeTx indicates the expression of TeTx protein.

**Figure supplement 3.** Reversal rates from an independent PQR::TeTx strain PSC229 indicate PQR is not involved in male's enhanced local search behavior.

and further verify the prediction by dissecting a circuit responsible for the enhanced local search behavior in males.

## Male *C. elegans* develops multiple mechanisms to increase the mating efficiency

The overwhelming majority (99.9%) of the *C. elegans* population consists of hermaphrodites, which prefer self-fertilization over mating (*Ebert and Bargmann, 2024*; *Garcia et al., 2007*; *Woodruff et al., 2014*; *Wu et al., 2023*). Therefore, the evolution pressure has driven the rare males to develop multiple mechanisms to increase the mating efficiency.

We find that the male's neural network contains about 25% (41/161) male-specific neurons, while the neural network in hermaphrodite only contains about 1.6% (2/122) hermaphrodite-specific neurons that regulates hermaphrodite-specific egg-laying behavior (*Emmons, 2018*). Structural analysis shows that male-specific neurons form strong connections within male neural networks, revealing the important roles of these neurons and setting the basis for males' prioritized sexual-related behavior. Our functional simulation analysis confirms the prediction based on structural features, emphasizing the substantial impact of male-specific neurons on the male neural network.

Not only the dominant role of male-specific neurons, the distinct connections and functions of sex-shared neurons facilitate sexual-related behaviors through different mechanisms. Males express pheromone receptors in both sex-shared sensory neurons (*Aprison and Ruvinsky, 2017*; *Fagan et al., 2018*) and male-specific neurons (*Narayan et al., 2016*; *Reilly et al., 2023*) to increase pheromone sensitivity. Besides the increased pheromone receptors in the sensory neurons, males rewire the synaptic connections at the interneuron and motor neuron level to promote sex-related behavior. AVA neuron is the command neuron for driving reversals, which plays an essential role for hermaphrodite behaviors, including navigation, food searching, aversive response, learning, and memory, etc. (*Choi et al., 2020*; *Gray et al., 2005*; *Liu et al., 2020*). We find that AVA has strong connections with sex-specific neurons and shows the key position for the functional connection in the whole male network as well as the male-specific sub-network. It suggests that AVA plays an important role for mating behavior in males. Indeed, a previous study demonstrated that AVA integrates the signals for sex-specific neurons to complete a series of stereotyped steps during mating (*Sherlekar et al., 2013*). Beyond directly regulating mating behavior, we find that the AVA-related rewired circuit in males mediates increased local search behavior that indirectly enhances the mating efficiency. The highly increased connections between command neuron AVA and downstream motor neurons VA and DA indicate a strong motor control in males. Sex-shared neurons DVC and RIC have increased chemical synapses with AVA in male, and our behavioral assay and calcium imaging results show that the circuit involving glutamatergic neuron DVC, octopaminergic neuron RIC, and command neuron AVA regulate enhanced spontaneous local search behavior that facilitates mate-searching. For further prioritizing the value of hermaphrodites, males decrease food attraction to promote food lawn leaving (*Ryan et al., 2014*), and decrease aversive response to nociceptive stimuli (*Pechuk et al., 2022*).

## Structural and dynamical feature-based sexually dimorphic behavior prediction

In *C. elegans*, pharyngeal neurons that govern the pharyngeal pumping mechanism for feeding are organized into a specialized sub-network (*Cook et al., 2020*). Previous research has demonstrated that signals from non-pharyngeal neurons can modulate feeding behaviors through regulating pharyngeal neurons in hermaphrodites (*Dallière et al., 2016*; *Trojanowski et al., 2016*). Structural and functional

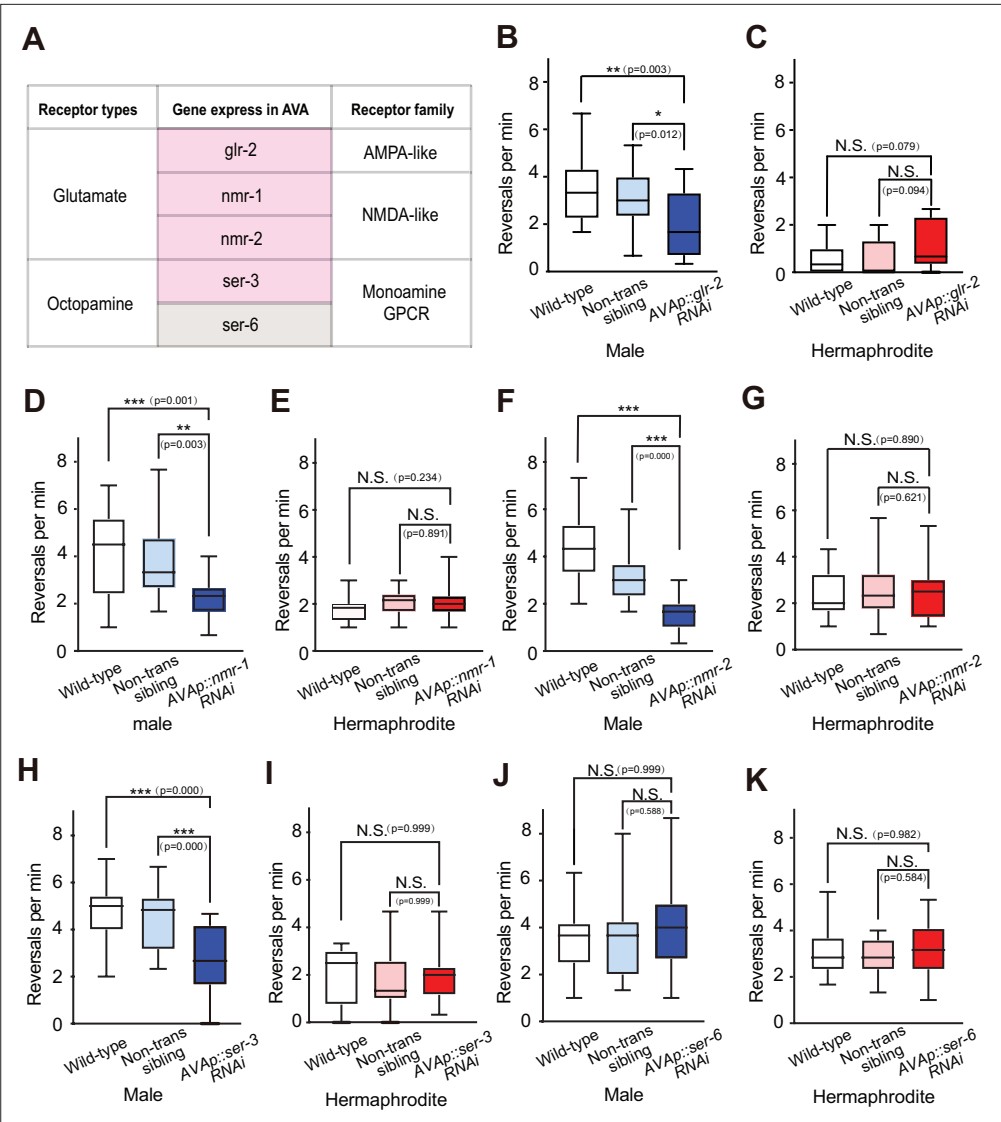

**Figure 7.** Receptors in AVA that are responsible for the increased local search in males. (**A**) Summary of receptor examined in this study. (**B**) Statistical testing for the average reversal rate between minutes 7–10 of 1-day adult males under different genetic backgrounds for glr-2 manipulation in AVA. 22 wild-type worms, 22 AVA::glr-2 RNAi transgenic worms, and 21 non-transgenic siblings. (**C**) Statistical testing for the average reversal rate between minutes 7–10 of 1-day adult hermaphrodites under different genetic backgrounds for glr-2 manipulation in AVA. 18 wild-type worms, 20 AVA::glr-2 RNAi transgenic worms, and 20 non-transgenic siblings. (**D**) Statistical testing for the average reversal rate between minutes 7–10 of 1-day adult males under different genetic backgrounds for nmr-1 manipulation in AVA. 20 wild-type worms, 19 AVA::nmr-1 RNAi transgenic worms, and 18 non-transgenic siblings. (**E**) Statistical testing for the average reversal rate between minutes 7–10 of 1-day adult hermaphrodites under different genetic backgrounds for nmr-1 manipulation in AVA. 24 wild-type worms, 20 AVA::nmr-1 RNAi transgenic worms, and 18 non-transgenic siblings. (**F**) Statistical testing for the average reversal rate between minutes 7–10 of 1-day adult males under different genetic backgrounds for nmr-2 manipulation in AVA. 19 wild-type worms, 19 AVA::nmr-2 RNAi transgenic worms, and 19 non-transgenic siblings. (**G**) Statistical testing for the average reversal rate between minutes 7–10 of 1-day adult hermaphrodites under different genetic backgrounds for nmr-1 manipulation in AVA. 20 wild-type worms, 20 AVA::nmr-2 RNAi transgenic worms, and 20 non-transgenic siblings. (**H**) Statistical testing for the average reversal rate between minutes 7–10 of 1-day adult males under different genetic backgrounds for ser-3 manipulation in AVA. 22 wild-type worms, 21 AVA::ser-3 RNAi transgenic worms, and 20 non-transgenic siblings. (**I**) Statistical testing for the average reversal rate between minutes 7–10 of 1-day adult hermaphrodites under different genetic backgrounds for ser-3 manipulation in AVA. 16 wild-type worms, 17 AVA::ser-3 RNAi transgenic worms, and 16 non-transgenic siblings. (**J**) Statistical testing for the average reversal

*Figure 7 continued on next page*

*Figure 7 continued*

rate between minutes 7–10 of 1-day adult males under different genetic backgrounds for ser-6 manipulation in AVA. 21 wild-type worms, 19 AVA::ser-6 RNAi transgenic worms, and 20 non-transgenic siblings. (**K**) Statistical testing for the average reversal rate between minutes 7–10 of 1-day adult hermaphrodites under different genetic backgrounds for ser-6 manipulation in AVA. 17 wild-type worms, 18 AVA::ser-6 RNAi transgenic worms, and 16 non-transgenic siblings. Student's *t*-test for all the statistical tests. *p<0.05, **p<0.01, ***p≤0.001.

The online version of this article includes the following figure supplement(s) for figure 7:

**Figure supplement 1.** Chemical synapse number difference of AVA with downstream neurons between male (**A**) and hermaphrodite (**B**).

---

studies characterized RIP neuron as a pivotal intermediary to connect pharyngeal neurons with others in hermaphrodite neural network (*Albertson and Thomson, 1976*; *Bhatla et al., 2015*). Interestingly, our current study reveals a distinct pattern in male worms. Unlike previous findings, we observed no direct connections between the pharyngeal neurons and other neurons, as depicted in *Figure 1—figure supplement 1*. Instead, we noted a robust interconnection among the pharynx neurons themselves and a loose functional linkage of the RIP neuron with sensory neurons responsible for internal and external environment changes (*Davis et al., 2018*; *Guo et al., 2018*; *Otarigho et al., 2024*; *Sato et al., 2021*) in neural network (*Figure 1N*). These findings suggest that males may possess an independent regulatory mechanism for feeding behaviors.

Our previous study demonstrated that the AIY→RIA circuit is fundamental for sensorimotor integration in olfactory steering in hermaphrodites (*Liu et al., 2022b*; *Liu et al., 2018a*). AIY receives inputs from various sensory neurons, including AWC and AWA, in both sexes (*Figure 1—figure supplement 4*). Notably, AWC and AWA in males are involved in sexual attraction (*Wan et al., 2019*; *White et al., 2007*). Here, we discover an enhanced circuit at the inter-motor neuron level, AIY→RIA→RMD_DV circuit (*Figure 1K and L*); therefore, we predict that enhanced AIY→RIA→RMD_DV circuit facilitates pheromone-guided olfactory steering behavior in males. A recent study showed that AFD expresses a dimorphic level of the insulin-like peptide INS-39 in males, suggesting a dimorphic function of AFD (*Haque et al., 2024*). Our analysis shows that neuron AFD exhibits the highest number of synaptic connections with AIY (*Figure 1—figure supplement 4*); therefore, further exploring the function of the AFD→AIY circuit in males is of significant interest.

Previous functional studies have demonstrated that AVA, as the command neuron, drives locomotion to regulate diverse behaviors in both sexes (*Bayer and Hobert, 2018*; *Choi et al., 2020*; *Gat et al., 2023*; *Gray et al., 2005*; *Lin et al., 2024*; *Pechuk et al., 2022*). Here, we present structural evidence for the essential role of AVA in the neural networks of males and hermaphrodites. We find that AVA has a large number of synapses, including both synaptic inputs and outputs, in two neural networks (*Figure 1B, G and K*). Interestingly, with respect to the difference in synapse numbers among sex-shared neurons, AVA exhibits a dimorphic connection pattern in males and hermaphrodites, as well as playing a key role in the sex-specific sub-neural network of males. Therefore, it is reasonable to predict that a wide range of behaviors mediated by AVA are sexually dimorphic. Here, we have demonstrated that AVA and the upstream neurons DVC and RIC regulate enhanced local search behavior in males (*Figures 5–7*). However, PQR, which reveals the largest synapse number difference with AVA between the two sexes, is not involved. Thus, it is intriguing to investigate the function of the PQR-AVA circuit in males to discover how this dimorphic circuit mediates sexually specific behavior.

We have identified several synaptic connections that exist only in one sex (*Supplementary file 3* and *Figure 3J*). One male-specific circuit involving PHA and two hermaphrodite-specific circuits comprising PHB and URY are highlighted according to the sex-specific connections (*Figure 2—figure supplement 2*). Indeed, studies have demonstrated the mechanisms of sex-specific pruning of PHA and PHB neuronal synapses and the corresponding sexually dimorphic behaviors (*Barrios et al., 2012*; *Oren-Suissa et al., 2016*; *Setty et al., 2022*). Sensory neuron URY is involved in pathogen avoidance in hermaphrodites (*Gallrein et al., 2021*; *Pujol et al., 2001*). Therefore, it is intriguing to examine whether URY activates downstream motor neurons RMD_DV and SMD or interneuron AVE, to generate avoidance behavior in a sex-specific manner.

Overall, this study elucidates the neural circuitry underlying prioritized sexual behaviors in males, also predicting potential dimorphic behaviors and the associated circuits.

## Materials and methods

### *C. elegans* strains

*C. elegans* strains maintenance followed the protocol from wormbook (*Stiernagle, 2006*). The Bristol N2 strain was used as wild-type. Males were generated through heat shock and maintained in a mixed population with hermaphrodites (*Anderson et al., 2010*). The following strains were employed in this study, with their detailed genotypes or microinjection mixtures described within square brackets:

1. GR1333, yzls71[Ptph-1::GFP +rol-6(su1006)] V;
2. PSC140, scnEx92[WY079(Pnmr-1sp::loxp::STOP::loxp::HisCl1), 30 ng/uL; XZ046(flp18p::Cre), 30 ng/uL; CC::GFP, 30 ng/uL];
3. PSC286, scnEx223[XZ046(flp18p::nCre), 31 ng/uL; YW379(nmr1sp::loxp::Stop::loxp::nGC6s), 6 ng/uL; CC::DsRed2, 39 ng/uL; pUC19, 24 ng/uL];
4. PSC138, scnEx90[YW397(Ptbh-1p::TeTx::mCherry), 30 ng/uL; CC::GFP, 31 ng/uL; pUC19, 30 ng/uL];
5. PSC412, scnEx302[YW398(ceh-63p::TeTx::mCherry), 45 ng/uL; CC::GFP, 30 ng/uL; pUC19, 30 ng/uL];
6. PSC227, scnEx166[YW396(gcy36p::TeTx::mCherry), 30 ng/uL; CC::GFP, 30 ng/uL; pUC19, 40 ng/uL];
7. PSC132, scnEx84[YW399(Pnmr1sp-nmr1 sense), 30.6 ng/uL; YW400(Pflp18p-nmr1_anti-sense), 30.4 ng/uL; CC::DsRed2, 39 ng/uL];
8. PSC136, scnEx88[YW401(Pnmr1sp-nmr2 sense), 30.1 ng/uL; YW402(Pflp18p-nmr2_anti-sense), 30 ng/uL; CC::DsRed2, 39 ng/uL];
9. PSC149, scnEx101[YW405(Pflp18p-glr2_anti-sense), 30.5 ng/uL; YW406(Pnmr1sp-glr2 sense), 30 ng/uL; CC::GFP, 31 ng/uL];
10. PSC233, scnEx172[YW474(Pnmr1sp-ser3-sense), 31 ng/uL; YW475(Pflp18p-ser3-anti sense), 30 ng/uL; CC::GFP, 40 ng/uL];
11. PSC320, scnEx245[YW495(Pnmr1sp-ser6 sense), 33 ng/uL; YW496(Pflp18p-ser6_anti-sense), 30 ng/uL; CC::GFP, 30 ng/uL];
12. PSC321, scnEx246[YW495(Pnmr1sp-ser6 sense), 33 ng/uL; YW496(Pflp18p-ser6_anti-sense), 30 ng/uL; CC::GFP, 30 ng/uL];
13. BNU001, scnEx223; scnEx90 [mated by PSC286 and PSC138];
14. BNU002, scnEx223; scnEx302 [mated by PSC286 and PSC412];
15. BNU003, scnEx223; scnEx166 [mated by PSC286 and PSC227].

### Molecular cloning of plasmids

The plasmids used to generate the aforementioned transgenic strains were constructed via Gibson assembly (*Gibson et al., 2009*), utilizing the Gibson Assembly Master Mix (NEB, E2611) according to the manufacturer's protocol. Gibson assembly is an isothermal, single-reaction method that assembles multiple overlapping DNA fragments through the concerted action of a 5' exonuclease, DNA polymerase, and DNA ligase. The DNA fragments (listed in *Supplementary file 5*) used for Gibson assembly were generated via PCR with the 2×High-Fidelity PCR Master Mix (MedChemExpress, Cat. No.: HY-K0533). Primer and template details are provided in *Supplementary file 5*. The identity of each plasmid was confirmed by sequencing.

For cell-specific expression, the following promoters were employed: AVA neuron expression: Driven by an overlapping strategy using the flp-18 promoter (Pflp-18, 1527 bp upstream of the flp-18 gene) and the nmr-1s promoter (Pnmr-1s, 990 bp upstream of the nmr-1s gene) *Zhang and Zhang, 2012*; RIC neuron-specific expression: Driven by the tbh-1 promoter (Ptbh-1, 2819 bp upstream of the tbh-1 gene) *Suo et al., 2006*; DVC neuron-specific expression: Driven by the ceh-63 promoter (Pceh-63, 646 bp upstream of the ceh-63 gene) *Oh et al., 2019*; PQR neuron expression: Driven by the gcy-36 promoter (Pgcy-26, 1089 bp upstream of the gcy-36 gene) (*Yu et al., 2017*).

### Cell-specific gene knockdown

Cell-specific gene knockdown was carried out based on the methodology outlined in a previous publication (*Esposito et al., 2007*) with some modifications. In this approach, double-stranded RNAs (dsRNAs), consisting of both sense and antisense RNA strands, are transcribed using cell-specific promoters. The DNA fragments responsible for generating the promoter::sense or promoter::antisense

RNA were subcloned into a plasmid, which was sequenced to confirm the accuracy of the dsRNA constructs.

For AVA neuron-specific RNA interference (RNAi), the expression of sense and antisense RNA strands was achieved using the aforementioned overlapping promoter strategy, where the flp-18 promoter drove the expression of the sense RNA, and the nmr-1s promoter drove the expression of the antisense RNA (*Zhang and Zhang, 2012*). This promoter-overlapping approach facilitated the generation of dsRNA specifically in AVA neurons. Details of the DNA fragments used to produce the dsRNAs are provided in *Supplementary file 5*.

Plasmids designed to express sense and antisense RNA were constructed using Gibson assembly. These plasmids, along with a microinjection marker plasmid (CC::GFP or CC::DsRed2), were injected into the germline of *C. elegans* to create RNAi transgenic lines, specifically PSC132, PSC136, PSC149, PSC233, PSC320, and PSC321.

## Graph theory analysis

In the graph theory analysis, several parameters were calculated. The definitions for each parameter are as follows:

1. Node strength: The node strength of a neuron is defined as the total number of synapses connected to that neuron.
2. Node out_strength: The node out_strength of a neuron is defined as the number of synaptic outputs from that neuron.
3. Node in_strength: The node in_strength of a neuron is defined as the number of synaptic inputs to that neuron.
4. Edge weight: The edge weight of a directed connection between two neurons is defined as the number of synapses comprising that connection.
5. Length of shortest path: The length of a path is defined as the reciprocal sum of the edge weights of the connections within the path. The shortest path between two neurons is the path with the minimum length.
6. Betweenness centrality: The betweenness centrality of a neuron is defined as the proportion of the number of shortest paths passing through that neuron relative to the total number of shortest paths in the entire neural network.

We generated an artificial network following the same principles as the neural networks in *C. elegans* (*Figure 1—figure supplement 5*) and used it as an example to demonstrate the calculation of each parameter.

1. Node strength: Neuron A: 2+3 + 3=8; Neuron B: 2+2 + 2+4 = 10; Neuron C: 3+1 + 2=6; Neuron D: 4+3 + 2+1 = 10.
2. Node out_strength: Neuron A: 2+3 = 5; Neuron B: 2+2 = 4; Neuron C: 3+1 = 4; Neuron D: 4.
3. Node in_strength: Neuron A: 3; Neuron B: 2+4 = 6; Neuron C: 2; Neuron D: 1+2 + 3=6.
4. Edge weight: $E_{A \to B} = 2$; $E_{A \to D} = 3$; $E_{B \to C} = 2$; $E_{B \to D} = 2$; $E_{C \to A} = 3$; $E_{C \to D} = 1$; $E_{D \to B} = 4$.
5. Length of shortest path: Here we take the pair of neurons B and D as an example. There are three possible paths from neuron B to neuron D, with the following lengths: $P_{B \to D} = \frac{1}{2}$; $P_{B \to C \to D} = \frac{1}{2} + 1 = \frac{3}{2}$; $P_{B \to C \to A \to D} = \frac{1}{2} + \frac{1}{3} + \frac{1}{3} = \frac{7}{6}$. Thus, the shortest path from neuron B to neuron D is the path B→D, with a length of 1/2.
6. Betweenness centrality: Total 12 shortest paths in the sample network are listed below with the lengths:

$$P_{A \to B} = \frac{1}{2}, \quad P_{A \to B \to C} = \frac{1}{2} + \frac{1}{2} = 1, \quad P_{A \to D} = \frac{1}{3};$$
$$P_{B \to C \to A} = \frac{1}{2} + \frac{1}{3} = \frac{5}{6}, \quad P_{B \to C} = \frac{1}{2}, \quad P_{B \to D} = \frac{1}{2};$$
$$P_{C \to A} = \frac{1}{3}, \quad P_{C \to A \to B} = \frac{1}{3} + \frac{1}{2} = \frac{5}{6}, \quad P_{C \to A \to D} = \frac{1}{3} + \frac{1}{3} = \frac{2}{3};$$
$$P_{D \to B \to C \to A} = \frac{1}{4} + \frac{1}{2} + \frac{1}{3} = \frac{13}{12}, \quad P_{D \to B} = \frac{1}{4}, \quad P_{D \to B \to C} = \frac{1}{4} + \frac{1}{2} = \frac{3}{4}.$$

Taking neuron D as an example, there are six shortest paths that pass through neuron D. Therefore, the betweenness centrality of neuron D is 6/12=0.5.

## Computational modeling

We model the dynamics of neural networks in both sexes of *Caenorhabditis elegans* using the RC neuron model. The RC neuron model is represented as follows:

$$c\frac{dV}{dt} = -g_{Na}(V - E_{Na}) - g_K(V - E_K) - g_L(V - E_L) + I_{ext}$$

In this equation, $V$ is the membrane potential, $c = 1.0$ is the membrane capacitance, $g_{Na} = 120.0$, $g_K = 36.0$, and $g_L = 0.03$ are the conductance of sodium ion channel, potassium ion channel, and leakage channel, respectively. $E_{Na} = 50.0\,mV$, $E_K = -77.0\,mV$, and $E_L = -54.4\,mV$ are the reversal potentials of the sodium ion, potassium ion, and leakage current, respectively. $I_{ext}$ is the external input current. A neuron receives synaptic currents from all upstream neurons, and the sum of all synaptic currents is its external input current $I_{ext}$. If there are $N$ synapses connecting neuron A to neuron B, then neuron B will receive a synaptic current $I_{A \to B}$,

$$I_{A \to B} = kN(V_A - V_B)$$

where $k = 100.0$ is the coefficient, $V_A$ and $V_B$ are the membrane potentials of neuron A and neuron B.

The details of the computation are provided in the Python code included in the supplementary materials (**Source code 1**).

## Local search behavioral assay

A 10 uL of OP50 bacterial liquid culture was transferred onto a 3 cm NGM plate and allowed to dry at room temperature for either 0.5 hours or 16 hours to differentiate food thickness. Individual one-day-old adult male or hermaphrodite *C. elegans* were then placed on the assay plate. The plate was positioned under a high-speed camera (MV-CA060-11GM, Hikvision) to record for 10 minutes at a frame rate of 10 Hz. Reversal behaviors were manually counted for further analysis. In the subsequent local search behavioral assay with transgenic animals, only the last 3 minutes of the 10-minute search period were recorded and analyzed.

## Manipulating AVA neuron with histamine administration

The protocol for temporally inhibiting AVA neurons with histamine treatment in AVA::hisCl1 transgenic animals was adapted from a previous study (**Pokala et al., 2014**). A 1 M histamine-dihydrochloride (H7125, Sigma-Aldrich) stock solution was prepared and stored in a –4°C freezer. For experimental use, the stock solution was diluted with double-distilled water (ddH$_2$O) to a final concentration of 200 mM. To prepare the histamine-containing food lawn, 10 uL of the 200 mM histamine-dihydrochloride solution was mixed with 190 uL of *E. coli* OP50 culture to achieve a final concentration of 10 mM histamine. Assay plates were prepared by seeding 10 uL of this bacterial mixture onto the center of 3 cm NGM plates and allowing the plates to dry for 0.5–3 hours before the experiment.

## Hermaphrodite preference assay

A 9 cm NGM assay plate pretreated with 10 uM histamine solution (H7125, Sigma-Aldrich) was divided into four equal sectors. A food lawn was created at the center of each sector by drying 50 uL of OP50 bacterial liquid culture for 2 hours at room temperature. Approximately 30 one-day-old adult GR1333 hermaphrodites were transferred onto two diagonal food lawns, initiating a 2-hour pheromone-releasing period. After this period, about 30 one-day-old adult males were quickly placed at the center of the assay plate and allowed to freely select a sector for 1 hour. Following the selection period, the number of males in each sector was manually counted. The Choice Index was calculated based on the distribution of males across the sectors:

$$Choice\ index(CI) = \frac{n(lawns\ with\ pheromone) - n(lawn\ without\ pheromone)}{n(lawns\ with\ pheromone) + n(lawn\ without\ pheromone)}$$

## Calcium imaging and data analysis

One-day-old males or hermaphrodites were paralyzed with 10 mM levamisole (A606644-0100, Sango Biotech) and placed on a glass slide with a thin layer of NGM agar. Images were captured for ten minutes at a frame rate of 4.84 frames per second using a Zeiss LSM980 confocal microscope equipped with an EMCCD camera and a ×60 oil immersion objective. The images were processed using ImageJ software. The dynamic tracking plugin "TurboReg" was used to pre-process the raw image gallery and correct any frame misalignment. Regions of interest (ROIs) were manually drawn around the AVA soma, and the gray values of these ROIs were measured to obtain the time series of

calcium signals. Approximately 144 frames (about 5% of the total 2876–2879 frames) were averaged to establish the $F_0$ baseline value. $\triangle F$ was calculated as $\triangle F = F - F_0$, where F represents the average fluorescence intensity of the ROI in each frame. The frequency of calcium events was determined by counting the number of signal intervals with $\triangle F/F_0$ exceeding a threshold value of 1 during the 10-minute recording period.

## Calcium imaging and data analysis

No explicit statistical power analysis was used to pre-determine sample sizes. Instead, sample sizes were chosen based on standard practices widely accepted in the *C. elegans* field to ensure adequate statistical power. All experiments were performed independently at least two to three times (for technical replicates on distinct days). Unless otherwise specified, the sample size (*n*) represents the number of independent biological replicates (e.g., independent synchronized worm populations or individual animals) rather than technical replicates. Age-synchronized populations of *C. elegans* were randomly allocated to experimental and control groups. Due to the obvious phenotypic differences or the nature of the experimental setup, investigators were not blinded to group allocation during data collection or analysis. Pre-established criteria for data exclusion were not required, and no outliers, data points, or animals were omitted from the final statistical analyses. All statistical analyses were performed using GraphPad Prism version 9.5. Data distribution was assumed to be normal, and variances were similar between groups being compared. Differences between two distinct groups were evaluated using an unpaired two-tailed Student's *t*-test (or Mann–Whitney U test for non-parametric data). For comparisons among three or more groups, a one-way analysis of variance (ANOVA) followed by Tukey's post hoc test was applied. Summary statistics are presented as mean ± standard error of the mean (SEM) In accordance with eLife guidelines, exact p values, exact sample sizes (*n*), and specific statistical tests used for each experiment are explicitly detailed in the respective figures.

## Acknowledgements

We thank Prof. Yun Zhang from Harvard University for generously providing plasmid XZ046 (flp18p::Cre), and plasmid WY079 (Pnmr-1sp::loxp::STOP::loxp::HisCl1) was generated during co-author Dr. Wenxing Yang's postdoctoral research in her lab. We acknowledge the Caenorhabditis Genetics Center for providing *C. elegans* strains. We also appreciate the technical support from the Instrumentation and Service Center for Science and Technology at Beijing Normal University. This work was supported by the National Natural Science Foundation of China (NSFC) under grant numbers 32371079, 32000720, and 32271178.

## Additional information

### Funding

| Funder | Grant reference number | Author |
| --- | --- | --- |
| National Natural Science Foundation of China | 32371079 | He Liu |
| National Natural Science Foundation of China | 32000720 | He Liu |
| National Natural Science Foundation of China | 32271178 | Wenxing Yang |

The funders had no role in study design, data collection and interpretation, or the decision to submit the work for publication.

### Author contributions

Xuebin Wang, Conceptualization, Investigation, Methodology, Project administration; Hanzhang Liu, Wenjing Yang, Conceptualization, Validation, Investigation, Visualization, Methodology, Project administration; Jingxuan Yang, Chunxiuzi Liu, Ke Zhang, Zengru Di, Wenxing Yang, Methodology; Xuehong Sun, Qiuhan Liu, Ying Zhu, Yinghao Sun, Investigation; Guiyuan Shi, Investigation, Methodology;

Qiang Liu, Visualization, Methodology; He Liu, Conceptualization, Supervision, Validation, Writing – original draft, Project administration, Writing – review and editing

**Author ORCIDs**
Xuebin Wang ⬚ https://orcid.org/0000-0002-3095-4417
Qiang Liu ⬚ https://orcid.org/0000-0002-9232-1420
He Liu ⬚ https://orcid.org/0000-0001-9418-9171

Reviewer #1 (Public review): https://doi.org/10.7554/eLife.102309.2.sa1
Reviewer #2 (Public review): https://doi.org/10.7554/eLife.102309.2.sa2

## Additional files

### Supplementary files
MDAR checklist

Supplementary file 1. The neural connection network of *Caenorhabditis elegans*.

Supplementary file 2. The functional classification of neurons with strong synaptic connections.

Supplementary file 3. Sex-specific synaptic connections.

Supplementary file 4. The most strongly correlated neuronal pairs in the stimulation simulation.

Supplementary file 5. The plasmids and reaction primers used in this study.

Source code 1. Neurodynamic simulation code.

### Data availability
All data supporting the findings of this study are available within the article and its Supplementary Information files.

The following previously published dataset was used:

| Author(s) | Year | Dataset title | Dataset URL | Database and Identifier |
|---|---|---|---|---|
| Cook SJ, Jarrell TA, Brittin CA, Wang Y, Bloniarz AE, Yakovlev MA, Nguyen KCQ, Tang LTH, Bayer EA, Duerr JS, Büow HZ, Hobert O, Hall DH, Emmons SW | 2019 | Adult hermaphrodite and adult male Data | https://wormwiring.org/pages/emmonslab.html | WormWiring, emmonslab |

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
