## [Editor Report · eLife Assessment]

This study presents **useful** findings on the differences between male and hermaphrodite *C. elegans* connectomes and how they may result in changes in locomotory behavioral outputs. However, the study appears **incomplete** with respect to the relationship between sex-specific AVA wiring and male mate-finding. Another area of concern is that the analysis does not consider animal-to-animal variability in the wiring when attempting to identify significant differences between the male and hermaphrodite.

---

## [Referee Report · Reviewer #1 (Public review)]

Summary:

This work seeks to predict differences in neural function and behavior between male and hermaphrodite *C. elegans* by comparing their nervous system maps of synaptic wiring. The authors then seek to validate some of their predictions by measuring differences in neural activity or behavior, including in response to neuron-specific genetic manipulations. In particular, the authors focus on the role of neuron AVA which has notable differences in its connectivity between the male and hermaphrodite, and they use this and behavior measurements to argue for a role of AVA in mate-searching behavior in males.

Strengths:

A major strength of this work is its approach to investigating differences in wiring between males and hermaphrodites in a systematic and quantitative way. The work laudably takes advantage of recently available comprehensive connectomes, including across sexes of the same species, and applies concepts from network science to mining their differences. Another strength of the work is that it supplements network analysis with measurements of behavior, including with cell-specific genetic manipulations. The measurements and analysis will be of value to the scientific community.

Weaknesses:

The evidence to support conclusions about the special relationship between differences in AVA's wiring and male mate-finding appears incomplete. The authors selected AVA based on changes in wiring and then observed a decrease in male chemotaxis towards hermaphrodites for animals in which neuron AVA is inhibited. This is presented as evidence that specifically AVA is important for mate-finding, and therefore that changes in wiring inform changes in function. But given AVA's known role in all reversal-related locomotion, it is important to more forcefully rule out an alternative hypothesis that the observed deficits in mate-finding could be explained by any reversal circuitry motor defect (including those without wiring differences), rather than specifically attributed to AVA and its wiring. Similarly, more evidence is needed to show that deficits in reversal circuitry preferentially affect mate-seeking compared to other goal-directed navigation behaviors.

There are some areas where methods would benefit from further justification or clarification. For example, the work would benefit from better justification for selecting sub-networks to study, or for combining bilaterally symmetric neurons. More details are also needed to better interpret calcium imaging studies, such as details about the indicator and illumination wavelength and intensity.

Finally, there are some weaknesses inherent to the entire field of connectomic analysis that are necessarily also present here. For example, it is unclear how to weight the relative contributions of chemical versus electrical gap junctions when performing analyses of the wiring diagram, and the choice could potentially influence results. The wiring diagram also lacks information about timescales of neural dynamics or the role of neuromodulators or other molecular details that may influence the strength or function of various connections, and this poses a major challenge for predicting neural dynamics from neural wiring. For example, in their neural dynamics simulation, the authors assume that all neurons have the same conductance and reversal potentials - a standard practice - despite known diversity among neurons that limits the usefulness of this approach. It will be helpful to further acknowledge these limitations of the broader field.

---

## [Referee Report · Reviewer #2 (Public review)]

Summary:

In their study, Wang and co-workers aimed to identify sexual dimorphisms in the connectomes of male and hermaphrodite *C. elegans*, and link these to sex-related behaviors. To this end they analyzed and compared various network properties of simplified male and hermaphrodite connectome datasets, and then focused on the AVA premotor neurons, linking their distinctive connectivity with their differential influence on reversing behaviors between the two sexes.

Strengths:

The study employs a range of basic methods from network and computational neuroscience and provides experimental testing of one of the predictions of the analysis.

Weaknesses:

Various aspects of sexual dimorphism in the nervous system of *C. elegans* have already been described and discussed (reviewed, for example, in Emmons 2018, Walsh et al. 2021). In particular, Cook et al, (2019), who mapped the male connectome (which serves as the key data in the current study), included in their work an analysis of connectome-level differences between males and hermaphrodites. Unfortunately, the foundations of the current study are somewhat problematic, and the results it provides are rather rudimentary and do not provide substantial new insight.

My critique of the study can be organized around several major issues.

(1) Source data

A large portion of the work is based on the analysis of a single male and a single hermaphrodite connectome datasets from Cook et al. 2019. These original connectomes were simplified in the current study, merging most individual neurons into neuron class nodes. As a measure of edge weight, the authors used the number of synaptic contacts between each two nodes. Cook et al. 2019 estimated this number to be of high variance, and even when considering unweighted connectivity (whether two nodes are at all connected or not) substantial variability exists between independent connectome datasets (e.g., Birari and Rabinowitch, 2024). Therefore, basing the analysis on synaptic weights from a single connectome (for each sex) may be somewhat unreliable.

On top of this, a huge gap may exist between connectome structure and function, especially when overlooking: (1) the sign of the synapses (excitatory vs. inhibitory), (2) synaptic efficiency (a single strong synapse may be more efficient than multiple weak synapses), (3) the spatial distribution of the synapses (clusters of synapses, for example, may be stronger than scattered synapses). These should at the very least be acknowledged. Moreover, the pooling of electrical and chemical synapses done by the authors is problematic, as is assuming all electrical synapses are bidirectional. These and other factors may undermine the results of the analysis, and, again, at the very least should be considered and discussed.

A minimal validation of the analysis could be achieved by sensitivity analyses. For example, studying how consistent the results are when: separately analyzing the chemical and electrical networks; binarizing synaptic contacts to existing vs. non-existing connections regardless of weight; and comparing with additional connectome datasets (at least for hermaphrodites).

Another important approach for validation would be synaptic labeling of key pathways, in order to establish the extent to which they maintain sexual dimorphism across the population (as performed, for example, by Cook et al., 2019; Pechuk et al. 2022).

(2) Statistical analysis

Comparing any two connectomes will show differences in connectivity and other network properties. The question is to what degree the differences found in the current study between two particular male and hermaphrodite connectomes transcend such basic inconsistencies. This fundamental question is not addressed in the manuscript.

A second major concern is that a considerable portion of the results are based on improper comparisons between male and hermaphrodite connectome measures.

In Figure 1D,I,M,V, Figure 2D,H,L, Figure 4E,I there is no sense in statistically testing the differences between hermaphrodite sex-specific (N=2) and shared nodes. The sample size is way too small. Corresponding conclusions about male-specific neurons being different from hermaphrodite-specific neurons in terms of connectivity are thus improperly founded. Similarly, the analyses in Figure 1P,S, 2O,R contain more data points, because of connectivity, but could still be misleading, since all the edges there contain either HSN or VC (just two nodes).

More so, any claim comparing the differences between two measures in males vs. hermaphrodites should be based on a 2X2 (or 3X2) design (e.g., tested using 2-way ANOVA with an interaction term). It is erroneous to interpret comparisons between two effects without directly comparing them (Makin et al., 2019).

When more than one comparison is performed, a one-way ANOVA should precede post hoc analyses, and corrections for multiple comparisons should be carried out and reported.

The plots in Figure 1E,W and Figure 4F,J are illustrative but do not contain any statistical test to support the claims about which functions are emphasized in which sex. They also rely on a very superficial categorization of individual neuron class function, whereas in reality, in *C. elegans* many neurons serve multiple functions.

In Figures 5-7 individual data points should be plotted, and the error bars and boxes should be defined (in all figures).

Finally, Figure 3C,F,I,L,N,P and Figure 5A-C lack statistical analysis (e.g., via bootstrapping). In addition, the term 'significantly' in the text should be reserved for statistical significance.

(3) Testing network predictions

A key emphasis of the network analysis concerns the AVA premotor neurons. It is well established that reversing behavior is controlled by premotor neurons such as AVA (e.g., Maricq et al. 1995) and that AVA activity is spontaneous and coupled to reversing (e.g., Chronis et al. 2007). More so, it has already been shown that male reversal frequency is higher than that of hermaphrodites (e.g., Mah et al. 1992; Zhao et al. 2003). Similar findings in the current study are thus not very surprising. The current study does add some new detail. Namely, the higher frequency of AVA activity in adult males compared to hermaphrodites, and the presumably sex-specific roles of RIC and DVC as well as several AVA glutamate receptors, in modulating reversing. At the same time, PQR, for example, showed no such role, contrary to the predictions.

Incidentally, AVA is not a commander neuron, but rather a command or, preferably, a premotor neuron. Altogether, the major specific focus of the analysis, predicting a sexually dimorphic role for AVA, is not very novel.

(4) Further predictions

The discussion section presents several additional predictions stemming from the analysis. However, to me, they seem almost arbitrary.

The statement claiming that the authors found the male pharyngeal connectome to be more strongly wired to the main connectome as opposed to previous findings, is unclear. Sex-specific differences in connectivity between the pharyngeal and somatic networks are immediately evident from the connectomes and do not require graph theoretical tools to be discovered (page 4 and discussion of Figure 3N).

The prediction that the AIY→RIA→RMD_DV circuit may facilitate pheromone-guided olfactory steering behavior in males is not very strong. On the one hand, it is known that males respond to sex pheromones (notably, however, if these pheromone receptors are ectopically expressed in hermaphrodites then hermaphrodites also respond to the pheromones [Wan et al. 2019]). Since these pheromone-sensing neurons are also involved in other sensory processes, it is quite trivial that the circuits involved in general sensory-based steering should be shared with specific pheromone-based steering. The fact that the interneurons in the circuit may be more strongly connected (excitatory, inhibitory, electrical?) in males could imply many things but does not add much to the picture.

The authors also mention AFD as having more synaptic contacts with AIY in males, and link this somehow to the dimorphic expression of insulin-like peptides in AFD. However, neuropeptide-based transmission is largely independent of synaptic connections, so I don't see the relevance.

(5) Methods

The example provided in the Methods section for calculating graph measures is very helpful. I am not sure, however, why the length of a path was defined as the reciprocal sum of the edge weights of the connections within the path. Why the reciprocal? Is it the sum of the reciprocals? Do more synaptic contacts imply a shorter path?

The description in the text (as opposed to the Methods section) of node strength is not very clear: "The node strength measures how strongly a node directly possesses with other nodes in the network" - This should be clarified.

For the RC simulation, I assume the sodium and potassium conductances are fixed. If so, they are leak currents themselves. What does the extra leak current represent? Obviously the simulation includes multiple arbitrary assumptions and parameter values. It would be useful to discuss at least the considerations for choosing the model design and parameters. I also assume that the delayed responses in the bottom neurons in Figure 4A (that still respond) are due to indirect synaptic connections (path lengths > 1)?